

# Evaluating the efficacy of bivariate extreme modelling approaches for multi-hazard scenarios

Aloïs Tilloy[1], Bruce D Malamud[1], Hugo Winter[2], Amélie Joly-Laugel[2]

[1]Department of Geography, King's College London, London WC2B 4BG, United Kingdom
5   [2]EDF Energy R&D UK Centre, Croydon CR0 2AJ, United Kingdom

*Correspondence to: Aloïs Tilloy (alois.tilloy@kcl.ac.uk)

Estimating risks generated by multi-hazard scenarios remains a challenge for practitioners. Here we evaluate the efficacy of bivariate extreme modelling approaches by fitting six distinct stochastic models to synthetic datasets. The properties of the synthetic datasets (marginal distributions, tail dependence structure) are chosen to match bivariate time series of environmental

10   variables. The six models are copulas (one non-parametric, one semi-parametric, four parametric). We build 60 distinct synthetic datasets based on different parameters of log-normal margins and two different copulas. We contrast the model strengths (model flexibility) and weaknesses (poorer fits to the data). We find that no one model fits our synthetic data for all parameters, but rather a range of models are more appropriate To highlight the benefits of the systematic modelling framework developed, we consider the following environmental data: (i) daily precipitation and maximum wind gust in London, UK; (ii)

15   daily mean temperature and wildfire number in Porto district, Portugal. In both cases there is good agreement in the estimation of bivariate return periods between models selected from the systematic framework developed in this study. Within this framework, we have explored a way to model multi-hazard events and identify the most efficient models for a given set of synthetic data and hazard sets.



## 1 Introduction

A multi-hazard approach considers more than one hazard in a given place and the interrelations between these hazard (Gill and Malamud, 2014). Multi-hazard events have the potential to cause damage to infrastructure and people that may differ greatly from the associated risks posed by a single hazard (Terzi et al., 2019). Here, we consider two main mechanisms in natural hazards interrelations defined in Tilloy et al. (2019): (i) cascade interrelations (i.e., when there is a temporal order and causality between natural hazards); (ii) compound interrelations (i.e., when several natural hazards are statistically dependent without causality).

Meteorological phenomena such as mid-latitude cyclones or thunderstorms often influence cascading and compound interrelations between natural hazards (e.g., a storm can include rain, lightening, hail, with rain and hail both potentially triggering landslides). Case examples of meteorological phenomena influencing natural hazard interrelations include the following:

(i)   In 2010, storm Xynthia hit the west coast of France. The storm itself was not particularly extreme for the season but the compound effect of extreme wind, high tides, storm surge, extreme rainfall and the fact that the soils were already saturated led to huge damage due to wind and flooding (CCR, 2019).

(ii)  In summer 2010, Russia experienced a heatwave. Low precipitation in spring 2010 led to a summer drought that contributed to the heatwave having a large magnitude (Barriopedro et al., 2011; Hauser et al., 2015; Zscheischler et al., 2018). The co-occurrence of extremely dry and hot conditions resulted in widespread wildfires, which damaged crops and caused human mortality (Barriopedro et al., 2011).

(iii) Extreme thunderstorms occurred in the Paris region in 2001, involving lightning and extreme rainfall, with the rainfall triggering flooding, mudslides and ground collapse, with subsequent damage to railway networks (CCR, 2019).


In this context, the quantification of interrelations between natural hazards can play an important role in risk mitigation and disaster risk reduction. Some of the natural hazards presented in the above examples are extreme occurrence of environmental variables (e.g. extreme temperature) which have different characteristics and statistical distributions (e.g., wind and landslides). Natural hazards are interrelated with different mechanisms (i.e. compound, cascade). For a given mechanism, interrelations also vary in strength and intensity. Additionally, as highlighted in Tilloy et al. (2019), different modelling approaches have been developed to quantify interrelations between variables. Here we focus on stochastic models that include copulas (Nelsen, 2006; Genest and Favre, 2007; Salvadori et al., 2016), and multivariate models (Heffernan and Tawn, 2004), limiting our analysis to the bivariate case. The potential for misinterpretation of the dependence structure of two variables clearly presents a problem when end-users try to account for hazard interrelations.


We choose six distinct bivariate models able to handle different types of tail (extreme) dependence: one non-parametric (JT-KDE), one semi-parametric (Cond-Ex) and four different parametric copulas (GalambosCop, Gumbelcop, FGMcop, Normalcop) (see **Sect. 2**). We are interested in modelling interrelations between hazards in the extreme domain. This implies the use of methods and concepts coming from the broad area of Extreme Value Theory (EVT). Amongst the six models compared in this study, four are directly linked to EVT (JT-KDE, Cond-Ex, GalambosCop, Gumbelcop). We investigate the applicability different bivariate models have to a wide range of natural hazard interrelations. In order to reproduce the complexity and variety of natural hazard interrelations we use 60 synthetic datasets to compare the fitting performances of the models. In these synthetics datasets we vary two main attributes of the bivariate datasets: the dependence structure and the marginal (individual) distributions. Of these 60 different synthetic datasets, 36 datasets have *asymptotically dependent variables* and 24 have *asymptotically independent variables* (see **Sect. 2.1** for a definition of these two concepts). The fitting capacities of each model are compared with the estimation of *level curves*. Level curves are extensively described in **Sect. 2.3**.



However, these curves correspond to probabilities that can be related to compound and cascading hazard interrelations. Compound interrelations are represented with a joint probability while cascading (sequential) interrelations are represented with conditional probabilities.


Examples of joint and conditional probabilities are given in **Fig. 1**. A joint probability is the probability of two events occurring together (**Fig. 1a**) and a conditional probability is the probability of an event given that another has already occurred (**Fig. 1b**).

**Figure 1** illustrates the concepts of joint probability and conditional probability, with daily rainfall data from a high-resolution gridded data set of daily climate over Europe (termed 'E-OBS') (Cornes et al., 2018) and daily maximum wind gust data at Heathrow airport provided by the Met Office (2019). A wind gust here is defined by the Met Office (2019) as the maximum 3 s average wind speed occurring in any given period. These datasets and the interrelation between extreme rainfall and extreme wind are discussed in **Sect. 4.1**.

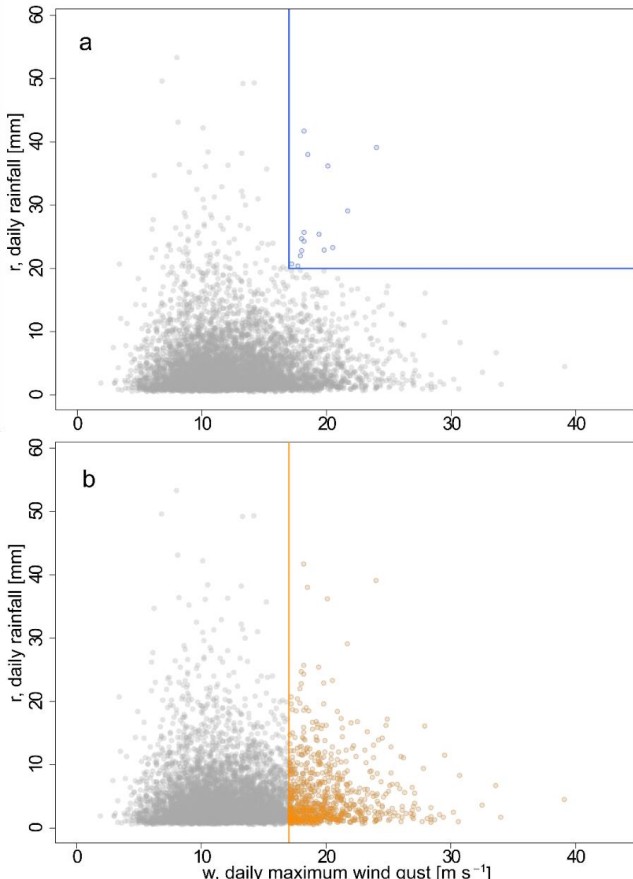


**Figure 1: Illustration of joint and conditional extremes with daily rainfall _r_ (mm day⁻¹) and daily maximum wind gust _w_ (m s⁻¹) data at Heathrow airport for the period 1971–2018: (a) joint extremes of rainfall and wind gust (blue circles); (b) conditional extremes of rainfall given that wind gust is extreme (yellow circles). Daily rainfall data from E-OBS (Cornes et al., 2018) and daily maximum wind gust (3 s period) data from the Met Office (2019).**

Joint and conditional probabilities are relevant metrics for practitioners and have been studied and used in several studies in the environmental sciences (e.g., Hao et al., 2017; Zscheischler and Seneviratne, 2017). However, as the most widely used level curve is the joint probability curve, we initially focus on it.

To analyse our results and compare the performances of the models we designed diagnostic tools that are presented in **Sect. 3.2**. The results of this study are presented as a map exhibiting the strength and weaknesses of our six models (**Sect. 3.3**). It

aims to provide objective criteria to justify the use of one model rather than another for a given set of hazards as it is shown with two examples in **Sect. 4**.

This article is organized as follows. We first (**Sect. 2**) provide a theoretical background on key concepts used in this study and present the models and methodology used. We then (**Sect. 3**) discuss the characteristics of our synthetic dataset and present




the results of simulation study. The diagnostic tools used to compare models are also discussed (i.e., joint return level curves

and dependence measure). Two applications (**Sect. 4**) to pairs of natural hazard data are presented: (i) compound daily rainfall

and wind in the United Kingdom and (ii) extreme hot temperatures and wildfires in Portugal. We finish (**Sect. 5**) with a

discussion and conclusions.

## 2 Methods

As stated in the introduction, we are interested in modelling the dependence structure between variables in the tails or extreme

domain. Extreme Value Theory has its roots in univariate studies (Coles, 2001) and has been extended to the multivariate

framework (Pickands, 1981; Davison and Huser, 2015). In this study we mainly focus on modelling the dependence between

two variables. Bivariate extreme value models developed within the statistical community (Resnick, 1987; Heffernan and

Tawn, 2004; Cooley et al., 2019) have recently been used for environmental application and therefore natural hazard

interrelations (De Haan and De Ronde, 1998; Zheng et al., 2014; Sadegh et al., 2017). A theoretical background on extreme

value theory is given in **Supplement S1.1**.

In this section, we first present the two types of asymptotic behaviour in bivariate extreme value statistics: asymptotic

dependence and asymptotic independence and discuss different dependence measures for the estimation of the relationship

between two variables (**Sect. 2.1**). The six bivariate models are then described (**Sect. 2.2**). Finally, we discuss the concept of

the return level in the bivariate framework (**Sect. 2.3**).

### 2.1 Bivariate extreme dependence

### 2.1.1 Asymptotic dependence and asymptotic independence

Let $X_1$, …, $X_n$ be $n$ different variables, with each variable a vector that can take on multiple values. Assume that these vectors

are random and independent and identically distributed (i.i.d). Then asymptotic dependence implies that if one variable $X_k$ for

$k \in (1, n)$ has values $X_k$ that are large it is possible for the other variables to take on values that are simultaneously extreme

(Coles et al., 1999). One way to characterise extremal dependence structures is to split them into those with asymptotic

dependence and those with asymptotic independence. In the bivariate case, for $(X_1, X_2)$ random pair with joint distribution $G$,

the random variables $X_1$ and $X_2$ are asymptotically dependent if the following conditional probability (Heffernan, 2000)

$$P(X_1 > x | X_2 > x) \to c > 0 \ as \ x \to x^* \tag{1}$$

Where $X_1 > x$ are those values of variable $X_1$ that are greater than a threshold $x$, the probability of both $X_1 > x$ and $X_2 > x$ is $c \in$

$(0,1]$ and $x^*$ is the upper end point (maximum) of the common marginal distribution.

The variables $X_1$ and $X_2$ are asymptotically independent if (Heffernan, 2000)

$$P(X_1 > x | X_2 > x) \to 0 \ as \ u \to x^* \tag{2}$$

where $u$ is a high threshold. In practice (Davison and Huser, 2015), extremal dependence is often observed to weaken at high

levels (i.e., as $u \to \infty$), and it can happen that dependence between variables is observed in the body of the joint distribution,

but that the multivariate distribution is in fact in the max-domain of attraction of independence (Davison and Huser, 2015).

Using models that take the assumption of asymptotic dependence (respectively independence) in the case of asymptotically

independent (respectively dependent) variables can lead to a large overestimation (respectively underestimation) of the

probability of joint extreme events (Ledford, 1996; Mazas and Hamm, 2017; Cooley et al., 2019).. Multivariate extreme value

and regular variation theory presented in the **Supplement S1.2** provides a rich theory for asymptotic dependence (De Haan

and Resnick, 1977; Pickands, 1981) but are not able to distinguish between asymptotic independence and full independence.





### 2.1.2 Tail dependence measures

A popular method to analyse hazard interrelationships is to compute dependence measures (Zheng et al., 2013; Petroliagkis, 2018). Dependence measures aim to describe how two (or more) variables are correlated.

When focusing on the dependence in the tails or extreme part of distributions, linear or rank dependence measures might not be accurate and other coefficients appear more relevant (Hao and Singh, 2016). Dependence between variables in the joint tail domain has been widely studied in the statistics community (Coles and Tawn, 1991; Ledford and Tawn, 1997; Coles et al.,

1999; Heffernan and Tawn, 2004; Zheng et al., 2014). As explained in **Sect. 2.1.1**, in the tails, two variables can be either asymptotically independent or asymptotically dependent; different diagnostics and coefficients previously developed are summarized in Heffernan (2000).

In this study, we use the following tail dependence measures:

- the extremal dependence measures $\chi$ and $\bar{\chi}$ introduced by Coles et al. (1999);
- the coefficient of tail dependence $\eta$, introduced by Ledford and Tawn (1996).

These coefficients aim to measure the extremal dependence for bivariate random variables $(X_1, X_2)$ and assume initially that $(X_1, X_2)$ have a common marginal distribution. Coles *et al.* (1999) defined the extremal dependence measure:

$$\chi(x) = P(X_2 > x | X_1 > x) \text{ with } \lim_{x \to x^*} \chi(x) = \chi \qquad (3)$$

with $x$ a sufficiently high threshold. A sufficiently high threshold $x$ is a value that can be considered as extreme within a given distribution (corresponding to a high quantile); the value of the threshold depends on the marginal distribution. The extremal

dependence measure $\chi(x)$ is the probability of one variable ($X_1$ or $X_2$) being extreme given the other is extreme ($X_2$ or $X_1$). This measure $\chi$ varies in the range [0,1], where a value of $\chi = 0$ means that the two variables are asymptotically independent and $\chi$ = 1 means that they are perfectly dependent. The extremal dependence measure $\chi$ is only suitable for asymptotic dependence. In the case of asymptotic independence ($\chi = 0$), Coles *et al.* (1999) introduced the measure $\bar{\chi}$ which falls between the range [-1,1], 1 being asymptotic independence. Ledford and Tawn (1996) defined their coefficient of tail dependence to be able to

assess the strength of dependence between two asymptotically independent variables. They show that the joint survivor function for random variables $(Z_1, Z_2)$ with common standard Fréchet margins (See **Supplement S1.2**):

$$P(Z_1 > z, Z_2 > z) \sim \mathcal{L}(z)(P(Z_1 > z))^{1/\eta} \qquad (4)$$

with $\mathcal{L}(z)$ a slowly varying function while $u \to \infty$ and $\eta$ is the coefficient of tail dependence, lying in the range [0,1]. Different values of each coefficient and their implications are summarized in **Fig. 2**. For large $z$, the three tail dependence measures presented above are related in the following way (Ledford and Tawn, 2003):

$$\bar{\chi} = 2\eta - 1 \qquad (5)$$

$$\chi = \begin{cases} 0 & \text{if } \bar{\chi} = 1 \text{ and } \mathcal{L}(z) \to c > 0 \text{ as } z \to z^* \\ & \text{if } \bar{\chi} < 1 \\ 1 & \end{cases}$$


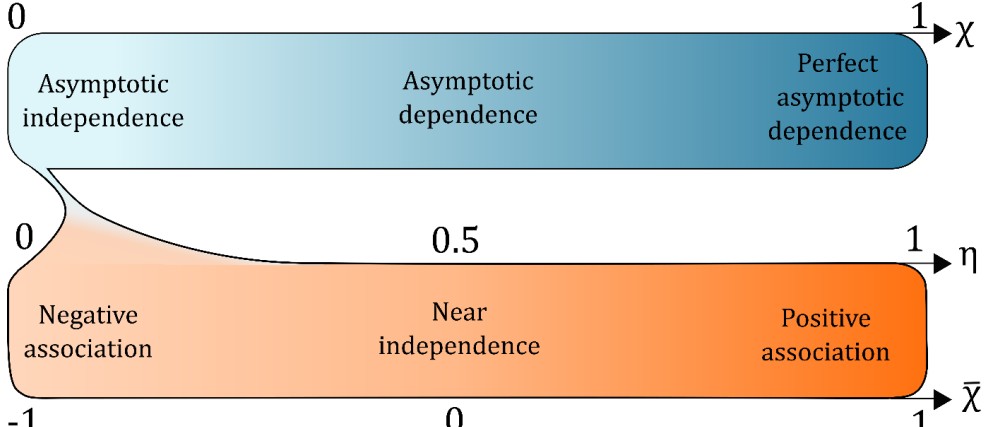

**Figure 2: The three coefficients used in this study to assess the dependence between two variables at an extreme level. In the upper part of the plot (blue), the coefficient $\chi$ varies between perfect asymptotic dependence (light blue, $\chi = 0$) and asymptotic independence (dark blue, $\chi = 1$). In the lower part of the plot (orange), which is in the asymptotic independence domain (in other words, $\chi = 0$) the coefficients $\bar{\chi}$ and $\eta$ both vary between negative association (light orange, $\bar{\chi} = -1$; $\eta = 0$) and positive association (dark orange, $\bar{\chi} = \eta = 1$).**

### 2.2 Bivariate models

Dependence measures are empirical measures which estimate the strength of the correlation, or dependence between two (or more) variables. Despite the fact that these measures provide crucial information, these do not allow to model joint (or conditional) exceedance probabilities. To model joint exceedance probabilities which represent the joint occurrence of hazards (here represented by extremes of environmental variables) in time and space, the use of stochastic models is required. In this section we present the three stochastic approaches for multivariate modelling that are used in the simulation study: parametric copulas, the semi-parametric conditional extremes model and a non-parametric approach based on multivariate extreme value theory (see **Supplement S1.2**) and kernel density estimation.

### 2.2.1 Copulas

In the bivariate case, a copula is a joint distribution function which defines the dependence between two variables independently from the marginal distributions of these variables (Heffernan, 2000; Nelsen, 2006; Genest and Favre, 2007; Hao and Singh, 2016). Let the random variables $(X_1, X_2)$ be vectors of i.i.d. values with marginal distributions $F_1(x_1)$ and $F_2(x_2)$ and a joint cumulative distribution function $F_{1,2}(x_1, x_2)$. Any bivariate distribution function with marginal distribution functions $F_{X1}(x_1)$ and $F_{X2}(x_2)$) can be expressed as a copula function as follows (Sklar, 1959; Nelsen, 2006):

$$F_1(x_1, x_2) = C\{F_1(x_1), F_2(x_2)\}, \qquad (6)$$

where $C$ is the copula function. Copulas are not limited to two variables and **Eq. 6** can be extended to higher dimensions. Several classes of copula with different properties are available, including Archimedean copulas, elliptical and extreme value copulas (Joe, 1997; Nelsen, 2006). Extreme value copulas have been used within various domains such as finance, insurance or hydrology because of their ability to model extremal dependence structures (Genest and Nešlehová, 2013).

However, extreme value copulas are by definition asymptotically dependent as they follow the rules of multivariate extreme value theory (see **Supplement S1.2**). The two types of extremal dependence were presented in **Sect. 2.1** and show that it is important to also consider asymptotic independence. Many copulas are asymptotically independent, including the normal copula and the Farlie-Gumbel-Morgenstern (FGM) copula (Heffernan, 2000). These two copulas will be used in the simulation analysis as asymptotically independent models (**Sect. 3**


Formally, the application of a copula model can be summarized in four main steps:

    (i)     Fitting marginal distributions to the two variables and then an empirical cumulative distribution function below a threshold and generalised Pareto distribution (GPD) above this threshold.

    (ii)    Transforming the variables to uniform margins. The transformed datasets no longer have information on the marginal distributions but keep the information about the joint distribution (Nelsen, 2006).

    (iii)   Fitting the copula function to the pseudo-observations by estimating the copula parameter(s) with an estimator (Genest and Favre, 2007).

    (iv)   Estimating the probability of joint events with the copula function previously fitted.

### 2.2.2 Conditional extreme model

The conditional extremes model (Heffernan and Tawn, 2004; Keef et al., 2013) is a semi-parametric model designed to overcome several limitations of copulas and other approaches such as the joint tail methods in which all variables must become large at the same rate. The aforementioned methods can typically handle only one form of extremal dependence, either asymptotic dependence or asymptotic independence. The conditional extremes model has the ability to be more flexible with asymptotic dependence classes; it can account for asymptotic independence and asymptotic dependence (Heffernan and Tawn, 2004; Keef et al., 2013). It can also be used to analyse more than two i.i.d variables more easily than copula-based methods (Winter and Tawn, 2016); we restrict the theory provided here to the bivariate case. The conditional model has been used for different purposes: spatial or temporal dependence between extremes (Winter and Tawn, 2016; Winter et al., 2016), dependence between extreme hazards (Zheng *et al.*, 2014) and even financial purposes (Hilal et al., 2011).

The conditional extremes model assesses the dependence structure between several variables conditioning on one being extreme and aims to model the conditional distribution. As in joint-tail models, the first step is to transform the marginal distributions; here the preferred marginal choice are Laplace (or Gumbel) margins (Heffernan and Tawn, 2004; Keef et al., 2013). Let the random variables $(Y_1, Y_2)$ be vectors of i.i.d. values with Laplace distributions. The conditional extremes model aims to identify normalizing two functions $a(y_i)$ and $b(y_i)$ such that $a$ satisfies $\mathbb{R}_+ \to \mathbb{R}$ and $b$ satisfies $\mathbb{R}_+ \to \mathbb{R}_+$, Both are defined such that for $y > 0$ (Winter, 2016):

$$P\left(\frac{Y_2 - a[Y_1]}{b[Y_1]} \leq z, Y_1 - u > y | Y_1 > u\right) \longrightarrow exp(-y)\, G(z) \tag{7}$$

as $u \to \infty$, where $G(z)$ is a non-degenerate distribution function. In the case of Laplace margins the normalising functions $a$ and $b$ are given by (Winter, 2016):

$$a[y] = \alpha y \quad and \quad b[y] = y^\beta \tag{8}$$

where $\alpha \in [-1, 1]$ and $\beta \in (-\infty, 1)$. The different values of $\alpha$ and $\beta$ characterise different forms of tail dependence. In the case where $\alpha = 1$ and $\beta = 0$, variables $(Y_1, Y_2)$ exhibit asymptotic positive dependence and the case of asymptotic negative dependence is given when $\alpha = -1$ and $\beta = 0$ (Winter, 2016).

Formally, the application of the conditional extreme model can be summarized in four main steps:

    (i)     Fitting marginal distributions to the two variables; an empirical cumulative distribution function below a threshold and generalised Pareto distribution (GPD) above this threshold.

    (ii)    Transforming those distributions onto Laplace (or Gumbel) margins.

    (iii)   Estimating the dependence parameters using non-linear regression.

    (iv)   Estimating the probability of joint events and extrapolate the conditional model to simulate new extreme data.



### 2.2.3 Joint tail KDE (kernel density estimation) approach

The non-parametric approach used in this paper is an adaptation of the non-parametric approach presented by Cooley et al. (2019). Moreover, the dependence measures $\eta$ is estimated to determine whether data are asymptotically dependent or asymptotically independent. This approach is based on the 2D kernel density estimator and the multivariate extreme value

framework (see **Supplement S1.2**).

The kernel density estimation (KDE) method has the advantage of being a non-parametric way to estimate the joint distribution of $n$ variables. With KDE, we make no assumption about the underlying distribution of the margins or about the dependence structure. The KDE centres a smooth kernel at each observation. The choice of the bandwidth is very crucial when using this method (Duong, 2007; Hao and Singh, 2016). This selection was done automatically in our case within the kernel survival

function estimation function from the R package ks (Duong, 2007, 2015).

The kernel density estimator is used here to estimate an empirical density distribution $\hat{f}(X)$ and a joint survival distribution $\hat{F}(X)$ of the bivariate dataset where $X=(X_1, X_2)$. The joint survival distribution corresponds to the joint exceedance probability of the two variables (See **Section 2.3**). From the joint survival distribution, it is possible to estimate level curves which are isolines corresponding to given joint probabilities of exceedance (see **Sect. 2.3**).

After estimating the joint survival distribution of the two variables with a kernel density estimator, the cumulative distributions $\hat{F}_i(x)$ of the two random variables $X_i$ ($i = 1, 2,…$) are estimated empirically below a threshold and from a Generalized Pareto distribution above the threshold. The two marginal cumulative distribution functions are then transformed to Fréchet margins to allow the use of multivariate extreme value theory(Cooley et al., 2019):

$$\hat{T}_i(x) = \frac{-1}{ln(\hat{F}_i(x))}. \tag{9}$$

Therefore, $Z = T(X) = \big(T_1(X_1), T_2(X_2)\big)$ can be assumed to be regularly varying with an index of regular variation 1 (see

**Supplement S1**). An extrapolation from a base probability $p_{\text{base}}$ (blue area in **Fig. 3**) estimated with a kernel density to an objective probability $p_{\text{obj}}$ (purple area in **Fig. 3**) is then done on the transformed space. Thus, on the transformed scale, it is possible to construct $\hat{l}_{Z(\text{obj})} = t\hat{l}_{Z(\text{base})}$ (Cooley et al., 2019). To produce level curves on the original scale, the transformation in **Eq. 11** is reversed: $\hat{l}_{\text{obj}} = T^{-1}\hat{l}_{Z(\text{obj})}$. **Figure 3** gives a graphical representation of the extrapolation done within the joint tail KDE approach.

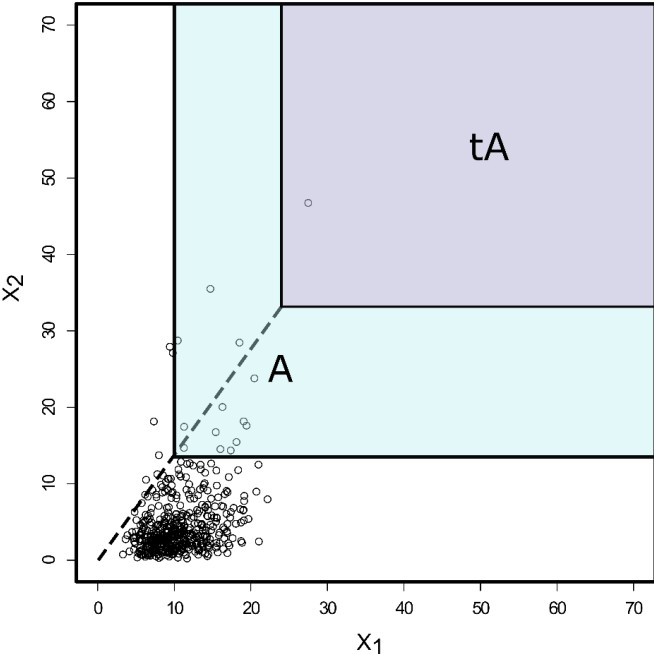


**Figure 3: Extrapolation in a regularly varying tail for a distribution in the max-domain of attraction of some multivariate extreme value distribution.** Black circles represent an asymptotically dependent bivariate dataset. In order to estimate the extreme joint probability P($tA$) (where $tA$ is an extreme set represented by the purple area), one can compute P($A$) = Pr{$Z \in A$}, (where A is a less extreme set than $tA$ represented by the light blue area) with $t < 1$. More data points are available in $A$ than $tA$, Then, from the regular variation framework
{$t$ P($tA$)} ≈ $t$P($A$). Adapted from Huser (2013)

The methodology presented above is only valid when the two variables $X_1$, $X_2$ are asymptotically dependent. In the asymptotic independence case, one needs to adjust the methodology. Two asymptotically independent variables follow the properties of hidden regular variation (Resnick, 2002; Maulik and Resnick, 2005) (see **Supplement S1.2.3**). Formally, the coefficient of tail dependence $\eta$ is introduced such as (Cooley et al., 2019):

$$\hat{l}_{Z(obj)} = t^{\frac{1}{\eta}}\hat{l}_{Z(base)} \qquad (10)$$

The specificity of this approach (presented below) is that it combines a non-parametric estimation of the joint density and the framework of multivariate extreme value presented in the **Supplement S1.2**. It can deal with both asymptotic dependence and independence. The coefficient of tail dependence has an influence on the extrapolation process in the asymptotic independence case. Here we used the one presented in Winter (2016) which is derived from the joint-tail model of Ledford and Tawn (1997). Formally, the application of the joint tail KDE model can be summarized in five main steps:

(i)    Estimating the joint cumulative distribution of the variables with a kernel density estimator.

       (ii)   Fitting marginal distributions to the two variables; empirical below a threshold and General Pareto Distribution (GPD) above this threshold.

       (iii)  Transforming those distributions onto Fréchet margins.

       (iv)   Determining whether variables are asymptotically dependent or asymptotically independent by estimating the
255           coefficient of tail dependence $\eta$.

       (v)    Estimating the probability of joint events and extrapolate the base isoline to an objective isoline.



### 2.2.4 Summary

In this section we have presented three modelling approaches: copulas, conditional extreme model and a non-parametric joint-tail model. All of these different approaches aim to model and reproduce the behaviour of multivariate data with different
dependence structures. In the next section we discuss the concept of return levels in the bivariate framework as an output measure of interest.

### 2.3 Return levels in the bivariate framework:

Studying natural hazards as multivariate — and particularly bivariate — events is a growing practice in multiple disciplines, including the following: coastal engineering (Hawkes et al., 2002; Mazas and Hamm, 2017); climatology Hao et al., 2017,
2018; Zscheischler and Seneviratne, 2017); and hydrology (Zheng et al., 2014; Hao and Singh, 2016). There has been debate among scientists trying to define a "multivariate return period" (Serinaldi, 2015; Gouldby et al., 2017). Serinaldi (2015) defined seven different types of probabilities that can be considered as bivariate probabilities of exceedance. These can be expressed through copula notation.

Let the random variables $(X_1, X_2)$ be vectors of i.i.d. values with marginal distributions $F_i(x_i)$ with i=1,2, $C$ their copula function
(**Sect. 2.3.1**) and $F_{1,2}(x_1, x_2) = C\{F_1(x_1), F_2(x_2)\} = C(u, v)$ where $F_{1,2}$ is the bivariate distribution function of $X_1$ and $X_2$, $u = F_1$ and $v = F_2$ are standard uniform random variables. The seven types of probability and their equations are given in **Table 1**.

Table 1: Types of probabilities for bivariate (X,Y) return period estimation. *u* and *v* are extreme thresholds. From (Serinaldi, 2015).

| Type of probability | Equation | Eq. # |
|---|---|---|
| $P_{AND}$ | $P(X > u \cap Y > v) = 1 - u - v + C(u, v)$ | (11) |
| $P_{OR}$ | $P(X > u \cup Y > v) = 1 - C(u, v)$ | (12) |
| $P_{COND1}$ | $P(X > u \mid Y > v) = (1 - u - v + C(u,v))/(1 - u)$ | (13) |
| $P_{COND2}$ | $P(X > u \mid Y \leq v) = 1 - \dfrac{C(u, v)}{u}$ | (14) |
| $P_{COND3}$ | $P(X > u \mid Y = v) = 1 - \dfrac{\partial C(u, v)}{\partial u}$ | (15) |
| $P_K$ | $P(C(u, v) > t) = 1 - K_C(t)$ | (16) |
| $P_S$ | $P(g(U, V)) = 1 - F_Z(z)$ | (17) |

The function $Kc$ in **Eq. 16** is the Kendall function and represents the distribution function of the copula (Salvadori and De Michele, 2010; Serinaldi, 2015). **Equation 17** refers to the 'structure-based' return period introduced by Volpi and Fiori (2014). Among these seven types of probabilities, we selected the "AND" and the "COND1" probabilities (see **Fig. 4**) as these are commonly used in the literature (Chebana and Ouarda, 2011; Tencer et al., 2014; Sadegh et al., 2018) and correspond to the two types of interrelations we are interested in (i.e., compound and cascade).

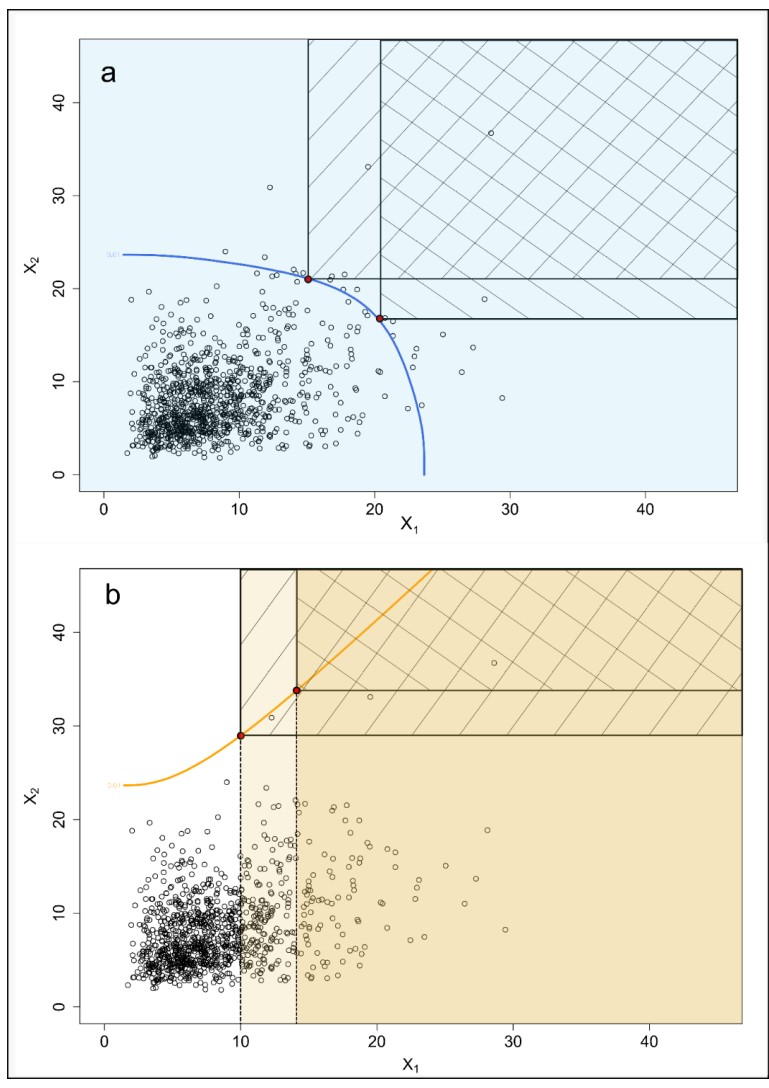

**Figure 4: Graphical representation of two bivariate ($X_1$, $X_2$) probabilities of exceedance: (a) $P_{AND}$ probability and (b) $P_{COND}$ probability with level curves (blue in 'a' and orange in 'b') representing $p = 0.01$ (1000 data points on a Gumbel copula with log-normal marginal distributions). Colours represent the domain on which the probabilities are computed while the areas with diagonal hatching represent the critical regions which are the regions corresponding to the given probabilities.**

In 2D space, probabilities of exceedance (or quantiles) are not represented by a single value but by a curve with an infinite amount of points with the same probability of exceedance. However, as shown in **Fig. 4**, these probabilities are defined by: (i) the domain where these are computed and (ii) the critical region corresponding to the probability type. For the AND probability, the computation domain remains similar when moving along the curve while the critical region evolves constantly. For the COND1 probability, both computation domain and critical region evolve when moving along the curve (see **Fig. 4**). Bivariate probabilities of exceedance are curves. These curves have been given various names in different research papers including the following:

- *isolines* (Salvadori, 2004; De Michele et al., 2007; Salvadori et al., 2016; Sadegh et al., 2017, 2018)
- *level curves* (Coles, 2001; Salvadori, 2004; De Michele et al., 2007; Volpi and Fiori, 2012; Serinaldi, 2015, 2016; Bevacqua et al., 2017).

For the specific case of the AND probability, the following names have been used:



- *joint exceedance curves* (Hawkes et al., 2002; Hawkes, 2008; Mazas and Hamm, 2017).
- *quantile curves* (De Haan and De Ronde, 1998; Chebana and Ouarda, 2011).

## 3 Simulation study

Here we are interested in comparing the abilities of six different models presented in **Sect. 2.3** to reproduce a given dependence
structure. Will create 60 different synthetic dataset types with varying marginal distributions and dependence structures. By
doing this, we aim to produce bivariate synthetic datasets comparable to the ones studied in bivariate hazard analysis (Zheng
et al., 2014; Hendry et al., 2019). This will allow us to confront the six models against the synthetic datasets, as a reference for
bivariate hazard interrelation analysis (**See Sect. 4**). The six models compared in this simulation study are:

   (i)      the conditional extremes model (Cond-Ex) (**Sect. 2.3.2**);
(ii)     the non-parametric joint-tail model (JT-KDE) (**Sect. 2.3.3**);
   (iii)    the Gumbel copula (Gumcop) (**Sect. 2.3.1**);
   (iv)     the normal copula (Normalcop) (**Sect. 2.3.1**);
   (v)      the Farlie-Gumbel-Morgenstern (FGMcop) copula (**Sect. 2.3.1**);
   (vi)     the Galambos copula (Galamboscop) (**Sect. 2.3.1**).
Among the four copulas used here, two are asymptotically dependent (Gumbel and Galambos) and two are asymptotically
independent (normal and FGM). A description of the six models is given in **Table 2**. **Table 2** synthetizes a range of information
about all the six models used in this simulation study including their type (nonparametric, semiparametric, parametric),
equation, parameter range (if there is a parameter) and asymptotic modelling domain. This latter information is important to
interpret the result of the simulation study in **Sect. 3.3**.


**Table 2: Description of the six statistical models compared in this article.** The description includes the model name and acronym (used thorough the article), type of model (parametric, semi-parametric, non-parametric), the mathematical description, the parameter range (where relevant) and the asymptotic modelling domain (AI for asymptotic independence and AD for asymptotic dependence)

| Model name | Model acronym | Model type | Mathematical description | Parameter range | Asymptotic modelling domain |
|---|---|---|---|---|---|
| Joint tail KDE | JT-KDE | Non-Parametric | $P(Z \in sA^*) \approx s^{-1}P(Z \in A^*)$ | | AD |
| | | Semi-parametric | $P(Z \in sA^*) \approx s^{-1/\eta}P(Z \in A^*)$ | $\eta \in [0,1]$ | AI |
| Conditional Extremes Model | Cond-Ex | Semi-Parametric | $P\left(\frac{Y_2 - a[Y_1]}{b[Y_1]} \le z \,,\, Y_1 - u > y \mid Y_1 > u\right) \to \exp(-y)\,G(z)$ <br> for y> 0, as u → ∞ where G(z) is a non-degenerate distribution function. | | AI and AD |
| Gumbel copula | GumCop | Parametric | $C(u,v) = \exp\left\{-\left[(-\ln(u))^\theta + -(\ln(v))^\theta\right]^{1/\theta}\right\}$ | $\theta \in [1,\infty)$ | AD |
| Normal copula | NormalCop | Parametric | $C(u,v) = \int_{-\infty}^{\Phi^{-1}(u)} \int_{-\infty}^{\Phi^{-1}(v)} \frac{1}{2\pi\sqrt{1-\theta^2}} \exp\left(\frac{2\theta xy - x^2 - y^2}{2(1-\theta^2)}\right) dxdy$ <br> With Φ(.) the standard Gaussian distribution function | $\theta \in [-1,1]$ | AI |
| FGM copula | FGMCop | Parametric | $C(u,v) = uv[1 + \theta(1-u)(1-v)]$ | $\theta \in [-1,1]$ | AI |
| Galambos copula | GalambosCop | Parametric | $C(u,v) = uv\exp\left\{-\left[(-\ln(u))^{-\theta} + -(\ln(v))^{-\theta}\right]^{-1/\theta}\right\}$ | $\theta \in [0,\infty)$ | AD |

In this section, we first describe and display the synthetic data that have been generated to conduct this study. We shall then

present the measures used in this study to compare the level curves and the dependence measures estimated from the six models presented in **Table 2**. Finally, results of the simulation will be displayed and analysed.

### 3.1 Synthetic data

Synthetic datasets are often used to confront or compare different statistical models (Chebana and Ouarda, 2011; Zheng et al., 2014; Cooley et al., 2019). Here we generated 60 bivariate synthetic datasets representative of environmental data such as

daily rainfall, daily wind gust and daily wildfire occurrences (see **Sect. 4**). The number of synthetic data points we use here have been fixed to 5000 for each dataset. For the asymptotic dependence case, 36 distinct datasets are generated from a Gumbel copula (see **Supplement S1.3.1**); for the asymptotic independence case, 24 datasets are generated from a normal copula (see **Supplement S1.3.2**). Each synthetic dataset set of parameters has been used to generate 100 realizations to produce confidence intervals.

The synthetic datasets are generated from two marginal distributions and a dependence model (i.e., copula). Both marginal distributions are log-normal; the log-normal distribution has been used (among others) for the modelling of a wide range of natural hazards, including wind, flood and rainfall (Malamud and Turcotte, 2006; Clare et al., 2016; Loukatou et al., 2018; Nguyen Sinh et al., 2019).

Random variables $x$ with a log-normal distribution are governed by two parameters: the location parameter $\mu$ and the shape

parameter $\sigma$ which correspond respectively to the mean and the standard deviation of $y$, the variable's natural logarithm, i.e., $y = \ln(x)$ (Aitchison, 1957). The parameter $\sigma$ influences the shape of the distribution and the heaviness of the tail and the dispersion of a log-normal distribution mostly depends on the shape parameter (Koopmans et al., 1964)





We can characterize log-normal distributions with the coefficient of variation $c_v$ which is the ratio of the standard deviation $s$ of the log-normally distributed variable $x$ to its nonzero mean $\bar{x}$ (Malamud and Turcotte, 1999):

$$c_v = \frac{s}{\bar{x}} \tag{18}$$

The standard deviation $s$ and the nonzero mean $\bar{x}$ are both related to the two parameters $\mu$ and $\sigma$ of the log-normal distribution (see **Table 3**). The distribution used in the simulation study, the parameters and the relationship between these parameters and the different tail dependence measures are summarised in **Table 3**.

Table 3: Marginal distributions and copula used for the synthetic dataset

| Distribution | Cumulative density function | Parameters | Parameters values |
|---|---|---|---|
| Log-normal distribution | $F(x) = \Phi\left(\frac{(\ln(x) - \mu)}{\sigma}\right)$ | $\mu, \sigma$ of $y$, where $y = \ln(x)$ <br> $\bar{x} = exp(\mu + \sigma^2/2)$ <br> $s = \sqrt{(exp(\sigma^2 - 1)exp(2\mu + \sigma^2)}$ <br> $c_v = s/\bar{x}$ | **A:** $c_v$ =0.25 <br> **B:** $c_v$ =0.53 <br> **C:** $c_v$ =2.91 |
| Gumbel copula | $C(u,v) = exp\left\{-\left[(-\ln(u))^\theta + -(\ln(v))^\theta\right]^{1/\theta}\right\}$ | $\theta = log_2(2 - \chi)$ | $\chi$ = 0.05, 0.10, 0.30, 0.50, 0.70, 0.90 |
| Normal copula | $C(u,v) = \int_{-\infty}^{\Phi^{-1}(u)} \int_{-\infty}^{\Phi^{-1}(v)} \frac{1}{2\pi\sqrt{1-\theta^2}} exp\left(\frac{2\theta xy - x^2 - y^2}{2(1-\theta^2)}\right) dxdy$ | $\theta = 2\eta - 1$ | $\eta$ = 0.25, 0.50, 0.75, 0.90 |

where $\Phi$ is the cumulative distribution function of the standard normal distribution

We use three different coefficient of variations $c_v = 0.25$ (labelled as **A** for the rest of this paper), 0.53 (labelled **B**) and 2.91 (labelled **C**) (See **Table 3**). The log-normal distribution **A** ($c_v = 0.25$) produces a distribution close to the normal distribution. The distribution **C** ($c_v = 0.2.91$) is a highly right-skewed distribution. The distribution **B** ($c_v = 0.53$) is intermediate skewness between **A** and **B**. In the bivariate context, there are six possible combinations of these distributions: **AA**, **AB**, **AC**, **BB**, **BC**, and **CC**.

The dependence structure is represented by a Gumbel copula in the case of asymptotic dependence (AD) and a normal copula in the case of asymptotic independence (AI) as no copula can be both asymptotically independent and asymptotically dependent (Heffernan, 2000; Coles, 2001). The Gumbel copula is an extreme value copula, asymptotically dependent (see **Supplement Eq. S19**). The Gumbel copula function only has one parameter $\theta$ which can be related the extremal dependence measure $\chi$. Here, we vary $\chi$ between 0.05 (very weak asymptotic dependence) and 0.9 (strong asymptotic dependence) (see **Fig. 5**). The Normal (or Gaussian) copula is asymptotically independent. Its unique parameter is related to the coefficient of tail dependence $\eta$ (Heffernan, 2000). We vary $\eta$ from $\eta = 0.25$ (negative sub-asymptotic dependence) to $\eta = 0.9$ (positive sub-asymptotic dependence) (see **Fig. 5**). In total, ten different dependence structures were simulated for each of the six combinations of marginal distributions. The 60 bivariate synthetic datasets used in this study are displayed in **Fig. 5**.



**Figure 5: The 60 different synthetic bivariate datasets used in our simulation study. On the *y*-axis: the dependence strength (a) $\chi$ (for asymptotic dependence) and (b) $\eta$ (for asymptotic independence), vary from slightly negative association to heavily dependent (see also Fig. 2). On the *x*-axis AA to CC represent the marginal distributions that are part of the bivariate distributions (see Table 3) with A, B, C representing log-normal distributions with different coefficient of variations $c_V$ (A: $c_V = 0.25$; B: $c_V = 0.47$; C: $c_V = 0.95$).**

To compare the fitting capabilities of the different models presented in **Sect. 2.3**, we vary several characteristics of the synthetic dataset:





(i) *The shape of the marginal distributions*. Natural hazards can exhibit very diverse statistical properties depending not only on their type but also on the location where they occur (Sachs et al., 2012).

(ii) *The strength of the dependence represented by the parameter of the copula function*. The type and strength of the relationship between natural hazards can vary within a broad range depending on the natural hazard studied or the location (Gill and Malamud, 2014; Martius et al., 2016). In order to take into account both AD and AI cases, the two parameters $\chi$ and $\eta$ (**Sect. 2.2**) are used.

### 3.2 Diagnostic tools

There are many diagnostic tools, to assess the goodness-of-fit of parametric bivariate models (Arnold and Emerson, 2011; Couasnon et al., 2018; Genest et al., 2009, 2011; Genest and Nešlehová, 2013; Sadegh et al., 2017). Amongst these, some of the most popular are the following:

- Cramer–von Mises statistic (Arnold, Taylor and Emerson, John, 2011);
- Kolmogorov-Smirnov test(Arnold, Taylor and Emerson, John, 2011);

- Akaike information criterion (AIC) (Akaike, 1974);
- Bayesian information criterion (BIC) (Schwarz, 1978).

These measures have been developed in a univariate framework and then extended to the bivariate framework. Genest (2009) proposed several approaches for Cramer–von Mises and Kolmogorov-Smirnov goodness-of-fit tests for copulas. There are two issues we faced using these measures for our study:

(i) these criteria are designed to fit on the dependence structure of the whole dataset and not the extreme dependence which can be different;

(ii) in our study we aim to compare parametric and non-parametric models.

To tackle the first issue, goodness-of-fit test have been developed for extreme value copulas (Genest et al., 2011). The latter issues is more complicated; each modelling approach has its own fitting methodologies, and although it is now possible to

compare copulas against each other (Sadegh et al., 2017; Couasnon et al., 2018), it is more difficult to compare copulas against semi-parametric or non-parametric models. The measures mentioned above are not suitable as they imply parametric distributions to be compared against observations (Stephens, 1970; Arnold, Taylor and Emerson, John, 2011). It is then not possible to compare the goodness-of-fit on the whole range of the data.

However, we are interested in fitting capabilities in the extremes. The models will then be compared on the estimation of two

attributes of the synthetic data detailed below:

(i) The $P_{\text{AND}}$ probability of exceedance (**Sect. 2.3**) represented by the level curve at $p = 0.001$.

(ii) The tail dependence measures $\chi$ and $\eta$ (**Sect. 2.2.2**).

We present here the diagnostic tools related to the level curve. The tools used to compare tails dependence measures can be found in **Appendix A**.Here we chose to compare our six models with respect to their ability to reproduce a reference level

curve $l_{\text{obj}}$ ('obj' is again used to indicate objective) which corresponds to an extreme joint probability $p = 0.001$. For each model $i$ a level curve $l_{\text{obj},i}$ is computed. Several methods and criteria have been used in the literature to compare level curves to a reference including comparing the curves with a vertical point-wise distances between the underlying curves (Chebana and Ouarda, 2011). This approach finds it limitation when level curves do not share the same $x$-axis coordinate ($X_1$ axis). In **Fig. 6** is presented our procedure for computation of the goodness-of-fit indicators (described in further detail below). In **Fig.**

**6** the example modelled and reference curves do not reach the same coordinate on the $X_1$ axis, making it impossible to compare these two level curves between $X_2$=0.0 and $X_2$=0.3. Cooley et al. (2019) divided level curves into two parts, comparing six $x$-axis coordinates on one part and six $y$-axis on the other part, to overcome the aforementioned limitation. Here we chose to use a consistent criterion all along the curves to evaluate the distance between each modelled curve and the reference curve. The four steps we use are the following:





(i)    Each modelled and reference level curve is normalized by dividing its coordinates by their maximum values. With
              that process, the curves are bounded in the [0,1] by [0,1] space. The different indicators are then computed in this
              normalized space.

       (ii)   Cartesian coordinates ($x,y$) of the modelled and reference level curves are transformed to polar coordinates ($\theta$, r).

       (iii)  Each modelled and reference level curve is discretized via linear interpolation into points. Each point corresponds to
415           an angle value (triangles and dots on the curves in **Fig. 6**).

       (iv)   Points from both the modelled and reference level curves with the same angle are coupled. Indicators are computed
              at each of the 80 couples of points (see **Fig. 6**).

The indicator designed in this study is derived from the distance between the two curves and are listed in **Table 4**:

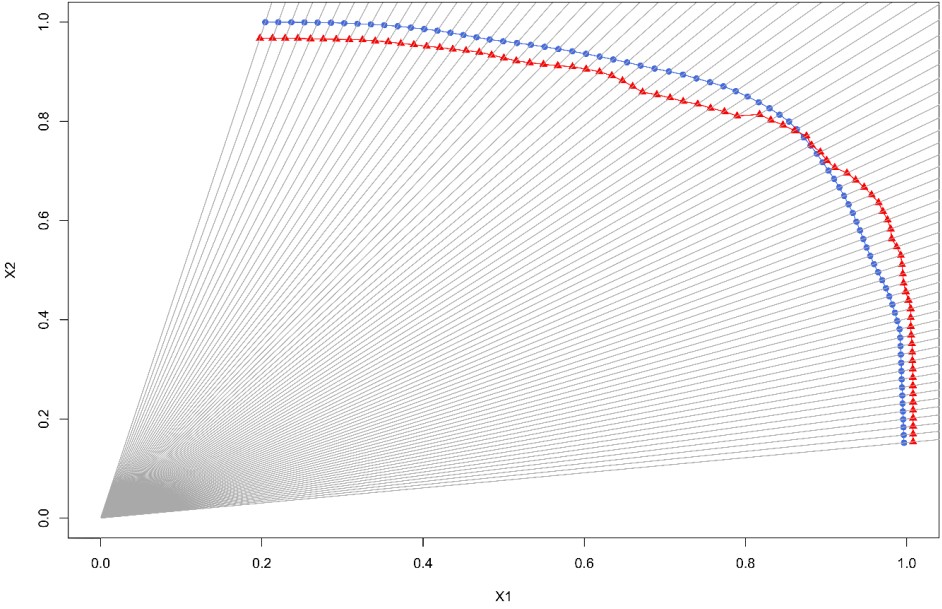

**Figure 6: Procedure for computation of the goodness-of-fit indicators.** Two variables are given, $X_2$ as a function of $X_1$. The
       red triangles and red curve represent the modelled level curve from a given model. The blue circles and blue curve are the
       reference level curve from the underlying bivariate ($X_1$, $X_2$) distribution of the data. Distance between the curves are calculated
       along the radius at 80 ($X_1$, $X_2$) coordinates (e.g., between the blue circles and the red triangles).

We used a weighted euclidean distance (wd) as comparison criteria. The density of level curves (described in the **Supplement
       S2**) allows one to weight the Euclidean distance of each of the 80 points by the local density of the curve. By weighting the
       Euclidean distance according to the reference bivariate distribution probability density function, we give more importance to
       the proper part of curve where a bivariate event is more likely to occur, rather than the naïve part (where the naïve part is
       defined as where the bivariate event is less likely to occur) (Chebana and Ouarda, 2011; Volpi and Fiori, 2012).

$$\sum_{i=1}^{N} w_i \left( \sqrt{(x_{mod,i} - x_{ref,i})^2 + (y_{mod,i} - y_{ref,i})^2} \right) \tag{19}$$

**3.3 Results**

       Two analyses are conducted in parallel, one for asymptotic dependence (AD) and one for asymptotic independence (AI). In
       the case of asymptotic dependence, the Gumbel copula is used with 5000 data points. The $\chi$ value is the one of interest under
       AD, values taken by $\chi$ have been presented in **Sect. 3.1**. For each $\chi$ value, we generated 100 realizations of the dataset from

the same underlying bivariate distribution. The 100 realizations generated have two purposes: (i) increase the robustness of

the results and (ii) create a confidence interval around the median which was set at the 95% confidence level by taking the quantiles Q2.5 and Q97.5 of the 100 realizations. To confront this approach, we generated two sets of 100 realizations which showed very small variations in the values of Q2.5, Q50 and Q97.5 without impacting our interpretation of the results. In the case of asymptotic independence, the normal copula is used. The $\eta$ value is the one of interest in the AI, values taken by $\eta$ are displayed in **Table 3**. For each $\eta$ value, we generated 100 realizations of the dataset. The 100 realizations generated allowed

us to create a 95% confidence interval using the same approach as above.

The marginal distributions do not have any impact on the dependence structure (Nelsen, 2006; Genest and Favre, 2007). We show in **Appendix A** that marginal distributions also have a very small impact on the estimation of dependence measures. All the methods used in this study include a transformation of marginal distributions and the fitting of a GPD above an extreme threshold (**Sect. 2.3**). By varying the marginal distribution of the variables of our synthetic dataset we aim to capture

uncertainties and errors arising from both the fitting of the marginal distributions and the dependence structure.

For both AD and AI, the objective level curve $l_{\mathrm{obj}}$ to be compared has been fixed at the probability $p_{\mathrm{obj}} = 0.001$. For each of the 60 bivariate datasets, the six models presented are fitted to the 100 realizations. The dependence measures $\hat{\chi}_i, \hat{\eta}_i$ as well as the level curve $\hat{l}_{obj,i}$ are estimated for each six model, with i∈ (1:6) correspond to each model. We then use the diagnostic tool and criteria presented in **Sect. 3.2** to compare the performance of the models. From the 100 realizations, 100 level curves

$\hat{l}_{obj,i}$ are generated for each model. Three curves are designed: (i) the 2.5% quantile level curve, (ii) The median level curve, (iii) the 97.5 % quantile level curve.

In an analogous way, for each of the diagnostic tools presented in **Sect. 3.2**, three values are computed: (i) the 2.5% quantile, (ii) the median, (iii) the 97.5% quantile. **Figure 7** displays the values of the *wd* for each model applied to each bivariate dataset. Squares are coloured according to the median of the *wd* and thickness of the edges is proportional to the size the confidence

interval (i.e., the distance between the quantiles Q2.5 and Q97.5).





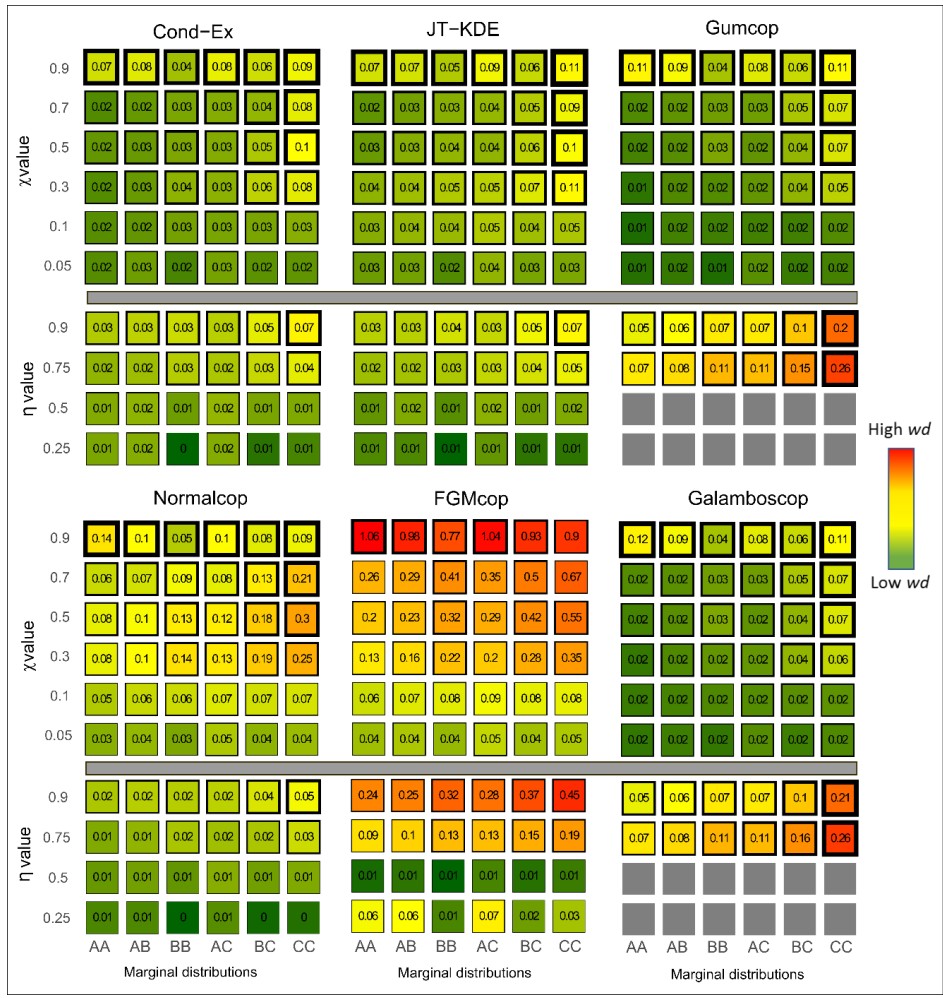

**Figure 7: Weighted Normalized Euclidean Distance (*wd*) to the reference curve for all 60 different synthetic datasets. Fitting capacities of each model are represented. Values in cells and colours represent the median *wd* from low (dark green) to high (red). Thickness of borders represent the 95% uncertainty around the median value on a logarithmic scale.**

It is important here to note that we tested more AD (36) cases than AI (24) cases. From **Fig. 7**, we observe the following:

- The Gumbel and normal copulas, which have been used to generate the synthetic datasets with AD and AI, outperform all the other models in AD and AI cases respectively.

- The conditional extremes model and the joint-tail KDE model are the most flexible model tested here as the can handle respectively 98% [72–100%] and 97% [65–100%] of the situation with a *wd* < 0.1, these values reach 100%

for the AI cases.

- The normal copula, even if asymptotically independent, is the most flexible copula model with a *wd* < 0.1 in 76% [60–90%] of the cases.

- Gumbel and Galambos copulas show very similar behaviours with respectively 68% [53–93%] and 68% [52–93%] cases with *wd* < 0.1. When only considering the 36 asymptotic independent cases, this value reaches 94%. It is

important to note that both aforementioned copulas cannot handle complete independence ($\eta$=0.5) or negative dependence ($\eta$ = 0.25).





- The FGM copula can only handle one type of extremal dependence, which is asymptotic independence with $\eta$=0.5. Consequently, it is the less flexible model in our results with a $wd < 0.1$ in only 41% [40–43%] of the cases.

- All models have poorer goodness-of-fit while shape parameters are increasing. It is particularly striking with the conditional extremes approach which exhibits high uncertainty and high $wd$ when both margins have a standard deviation $\sigma$=1.5.

The Cond-Ex and JT-KDE provide similar results according to **Fig. 7** despite adopting very different approaches. Thus, their flexibility arises from their semi parametric nature. **Figure 7** also displays the uncertainty of the estimate of $wd$. For all models, a more accurate fit is accompanied with a reduction in uncertainties. However, both Cond-Ex and JT-KDE have on average

more uncertainty around its $wd$ despite their good fitting capabilities. On average, copulas tend to have less uncertainty due to their parametric nature.

However, the copulas are penalized by the weighting function as they usually reproduce quite well the naïve part of the curve. By considering again the percentage of situations with a criterion below 0.1, the normal copula has its performances reduced by the weighting function ($-6\%$ compared to $d$). The JT-KDE model has its performance boosted by the weighting function

($+7\%$ compared to $d$).

**4 Application to natural hazards**

Results from the simulation study presented in the previous section (**Sect. 3**) can provide useful insights when modelling the interrelations between two natural hazards. In this section we will show how results previously presented can be useful to identify the most relevant models for a given dataset according to its visual characteristics. The concordance (or discordance)

of the relevant models can also increase (decrease) confidence around the results.

The methodology for model selection presented here is composed of five steps to select the most relevant models estimate joint exceedance probability level curves:

(i) A scatterplot is compared visually to density plots for the 60 different datasets simulated in **Sect. 3** and displayed in **Fig. 6**.

(ii) From the aforementioned 60 datasets, a set of one to six analogous datasets (i.e. with similar bivariate distribution) is taken.

(iii) A confidence score is used to compare the abilities of each models for the selected datasets in step (ii). For each model, the confidence score is $\overline{wd}$ the average of the computed weighted Euclidian distance $wd$ for all datasets selected in step (ii). By taking the average of $wd$, a poor fit on one analogous dataset will have a high influence on

the confidence score.

(iv) Models are fitted on the bivariate hazard dataset and level curves from the most relevant models are kept.

(v) Tail dependence measures are estimated using the most relevant model with a possible new iteration of the four previous steps according to the value of the dependence measures.

To produce a confidence interval as was done in the simulation study (**Sect. 3**) and to visually measure the uncertainty associated to each level curve as in **Sect. 3**, we use a nonparametric bootstrap procedure. The function *tsboot* from the R

package *boot* (Davison and Hinkley, 1997; Canty and Ripley, 2019) is used to generate 100 bootstrapped replicate datasets with the same number of observations as the original (but some are repeated). Our six models are then fitted to the original dataset and on the 100 bootstrapped replicates.

**4.1 Rain and wind gusts at Heathrow Airport (Asymptotic independence)**

Here we study the interrelation between daily extreme wind gusts ($w$) and extreme rainfall ($r$) at London Heathrow airport, UK for the period 1 January 1971 to 31 May 2018, both introduced in **Fig. 1**. The relationship between wind and rainfall has


been studied both globally (Martius et al., 2016) and locally (Johansson and Chen, 2003; Ming et al., 2015). These two hazards are often associated with different types of storms (Dowdy and Catto, 2017) and in particular cyclones (Ming et al., 2015; Raveh-Rubin and Wernli, 2016). In South England, these two hazards are mostly associated with mid-latitude cyclones in the

winter season and thunderstorms in summer season(Hawkes, 2008; Anderson and Klugmann, 2014; Webb and Elsom, 2016; Hendry et al., 2019).

The bivariate dataset used to study the interrelation between wing gusts and rainfall at Heathrow airport is composed of the following data:

a)    *Daily Wind Gust (w):* daily maximum wind gust at Heathrow airport (London, UK) weather station where a gust

is the maximum 3 second average of wind speed. Wind gusts are short lived wind peaks in speed that can inflict great damage during a storm; however it might not capture the overall wind intensity (Met Office, 2019). The time range of the observations is 38 years, from 1 January 1971 to 31 May 2018 of which 74 days (0.4% of the data) had no values recorded and all other values in the dataset had $w > 0$ m s$^{-1}$. This observation data has been provided by the Met Office (2019).

b)    *Daily Rainfall (r):* daily total precipitation in a grid cell containing Heathrow airport. The data have been extracted from the E-OBS gridded database (Cornes et al., 2018) which is formed from the interpolation of observations from 18,595 meteorological stations through Europe and the Mediterranean (including Heathrow airport station). It has been shown that E-OBS has excellent correlation with other high-resolution gridded datasets even if this correlation tends to decrease for extremes (Hofstra et al., 2009). However, by selecting a

grid containing a weather station we limit uncertainties arising from interpolation. The spatial resolution in the E-OBS dataset is 0.1° x 0.1° and the period covered is 1950 to 2019. Data from 1 January 1971 to 31 May 2018 (38 years) in the cell containing Heathrow airport is used, with a total of 6074 days (35.1% of the dataset) having nonzero rainfall $r > 0$ mm d$^{-1}$.

From 1 January 1971 to 31 May 2018 there are a total of 17,318 days (including leap years). Our bivariate wind gust-rainfall

dataset is composed of those values where there is both non-zero rainfall $r > 0$ mm d$^{-1}$ and windgusts $w > 0$ m s$^{-1}$ recorded, resulting in a total of 6044 bivariate observations (34.9% of the days in our record). An overview of both daily rainfall and daily wind gust is displayed in **Fig 8** in the form of monthly violin plots, where the probability density of *w* and *r* at different values are given, smoothed by a kernel density estimator.
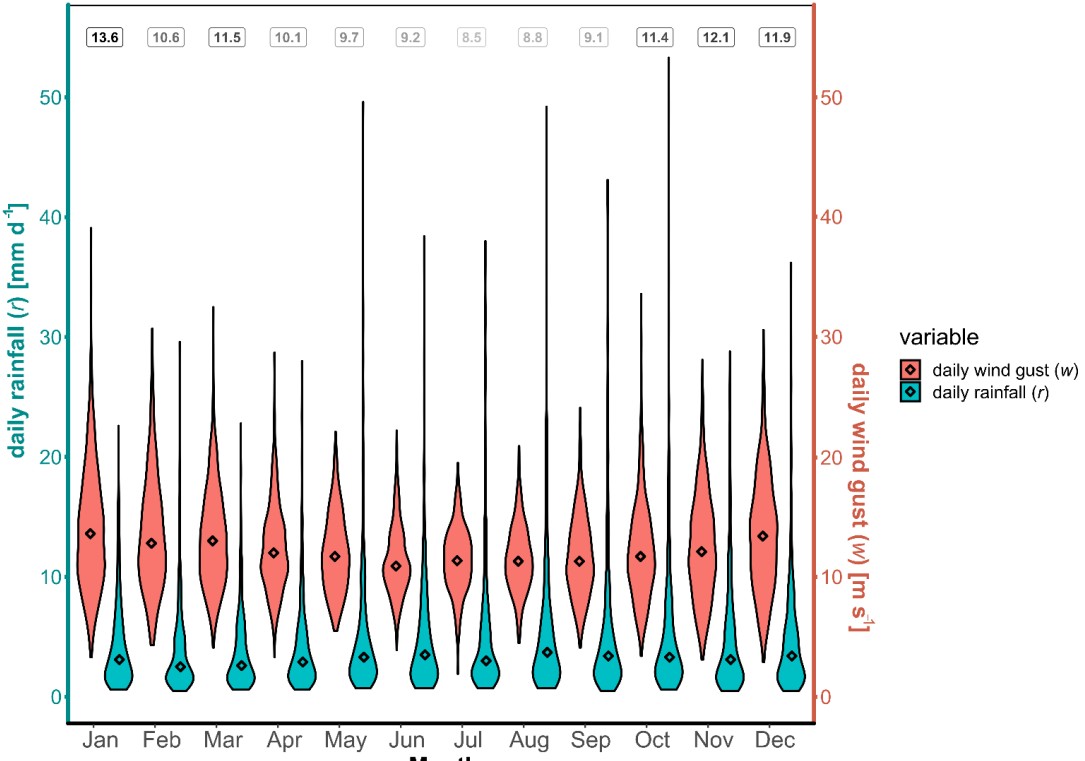

**Figure 8: Violin plots of daily wind gust $w$ (red) and daily non-zero rainfall $r$ (blue) by month for the period 1 January 1971 to 31 May 2018 at Heathrow airport weather station, UK. Diamonds represent the median of all values for that month from 1971-2018. Numbers at the top of the graph represent the average number of days per month where there is recorded both non-zero rainfall $r$ > 0 mm d$^{-1}$ and windgusts $w$ > 0 m s$^{-1}$. Daily rainfall data from E-OBS (Cornes et al., 2018) and wind gust data (maximum 3 s wind velocity in a day) from the Met Office (2019).**

From **Fig. 8** we observe a seasonality in daily wind gust speed. January is the month with the highest median (diamond symbol) and range of most values in the violin plot while July is the month with the lowest median and range of most values in the violin plot. The daily non-zero rainfall median per month varies between 2.5 mm in February and 3.5 mm in June, with the highest individual daily values occurring in October (53.3 mm d$^{-1}$), May (49.6 mm d$^{-1}$) and June (49.2 mm d$^{-1}$). The dataset is also represented as a scatterplot in **Fig. 9**. The scatterplot will be used for the model selection methodology presented at the beginning of **Section 4.**





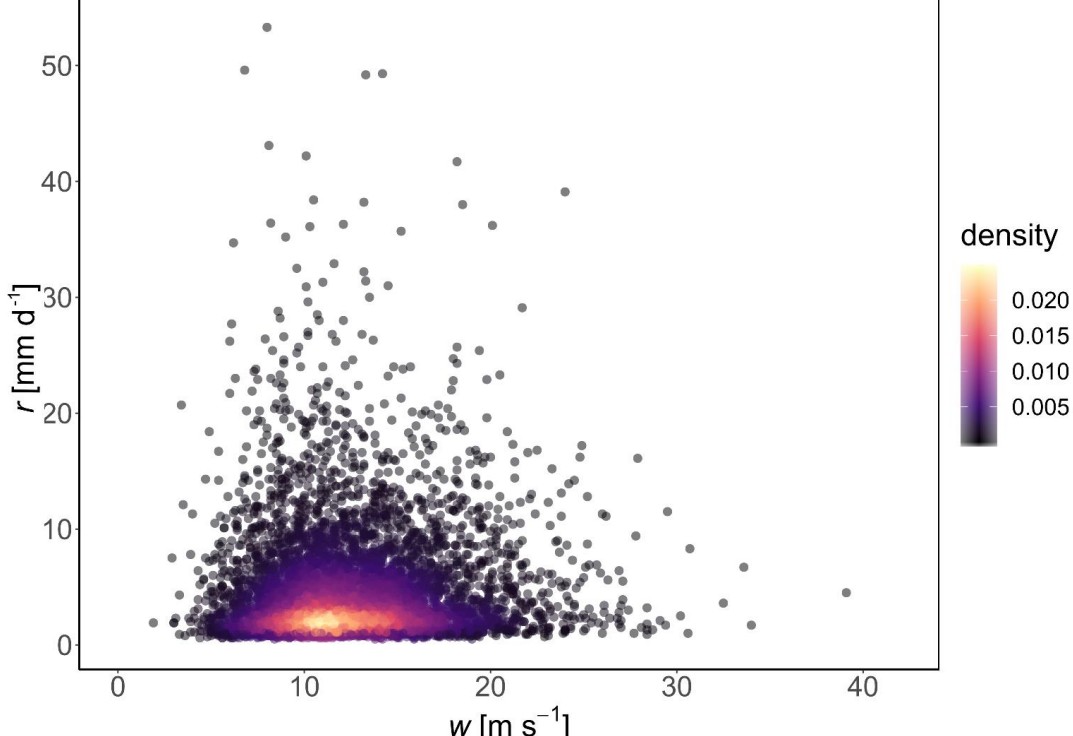

**Figure 9: Days where there are recorded both daily wind gust (m s⁻¹) $w$ and nonzero daily rainfall (mm d⁻¹) $r > 0$ mm d⁻¹ at Heathrow airport (London, UK) for the period 1971–2018. Daily rainfall data from E-OBS (Cornes et al., 2018) and wind gust data (the maximum 3 s wind velocity in a day) from the Met Office (2019).** Colours (legend) represent the bivariate density estimated from a kernel density estimator with higher values and lighter colours representing a higher density of points at that bivariate value ($r$, $w$).

Extreme rainfall and extreme wind have a compound interrelation according to Tilloy et al. (2019). We then estimate the joint exceedance probability curve, corresponding to a P$_{AND}$ probability (**Sect. 2.3**).

We now go through the four steps presented for rainfall and wind gusts in Heathrow.

(i)    From **Figs. 9** and **6**, we hypothesize that over our time range 1971-2018, daily rainfall and daily maximum wind gusts in Heathrow are asymptotically independent or weakly dependent ($\eta = 0.5$ / $\chi = 0.05$) and that both marginal distributions have a small shape parameter (A-B, B-B).

(ii)   This then gives us four analogous datasets and it is then possible to visually infer from **Fig. 6** which models are the most adapted to these conditions. The four analogous datasets are the following:

- $\chi = 0.05$ and A-B
- $\chi = 0.05$ and B-B
- $\eta = 0.5$ and A-B
- $\eta = 0.5$ and B-B

(iii)  The *confidence score* for each model is $\overline{wd}$ the average of the weighted Euclidean distance $wd$ from the four situations above. For the Gumbel and Galambos copulas, the cases of independence or negative dependence between variables are outside the modelling range (**Sect. 2.3.1**), and thus the confidence score for these models has been penalized by putting $wd = 1.0$ for $\eta = 0.5$ and $\eta = 0.25$. The conditional extremes model, normal copula and JT-KDE model all have a confidence score variable = 0.02. The FGM copula has a confidence score of variable = 0.03. Gumbel and Galambos copulas have a confidence score of variable = 0.5 due to their penalty (**Table 4**).





According to these three first steps, we can identify the four most relevant model for the bivariate dataset of daily rainfall and daily wind gust at Heathrow. The conditional extreme model, the JT-KDE model, the normal copula and the FGM copula are the most relevant models for our dataset as can be seen in **Table 5**.

**Table 4: Euclidian weighted distance (*wd*) for datasets 1 to 4 based on wind-rainfall and six models, along with confidence scores (average of the wd for datasts 1 to 4). In bold are highlighted the four models with confidence scores < 0.1.**

| Dataset | Cond-Ex | JT-KDE | Gumbelcop | Normalcop | FGMcop | GalambosCop |
|---|---|---|---|---|---|---|
| 1 | 0.03 | 0.03 | 0.02 | 0.04 | 0.04 | 0.02 |
| 2 | 0.02 | 0.02 | 0.01 | 0.03 | 0.04 | 0.02 |
| 3 | 0.02 | 0.02 | 1.00 | 0.01 | 0.01 | 1.00 |
| 4 | 0.01 | 0.01 | 1.00 | 0.01 | 0.01 | 1.00 |
| Average | **0.02** | **0.02** | 0.51 | **0.02** | **0.03** | 0.51 |

(iv)     For illustration and/or confronting our models with the data, the six models are fit to the dataset and joint exceedance level cure are produced with a joint exceedance probability set at $p = 0.001$, corresponding to a bivariate return period of 8 years. However, another joint exceedance probability could have been chosen.

In **Fig. 10** are displayed the level curves produced from the four models that were selected after steps (i) to (iii) above (Cond-Ex, JT-KDE, NormalCop and FGMCop) and presented in bold numbers in **Table 4**.

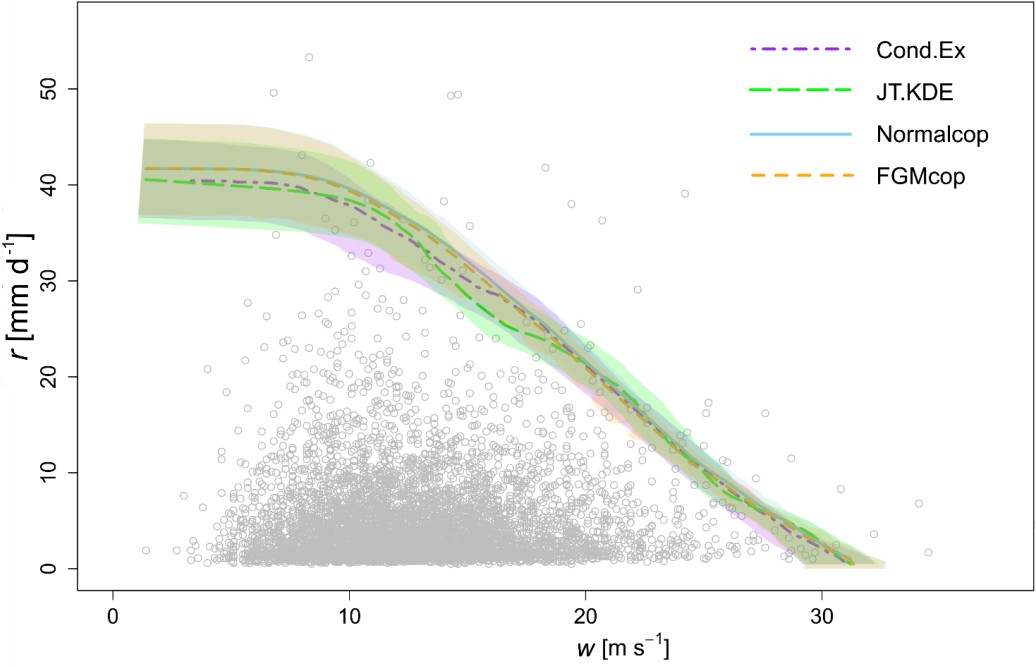

**Figure 10: Level curves for a $P_{and}$ joint probability $p = 0.001$ of daily wind gust and daily rainfall at Heathrow airport (London, UK). Level curves from the four models selected through the model selection methodology are displayed.**

From **Fig. 10,** we can observe that the conditional extremes model, the FGM and the normal copula all produce very similar joint exceedance curve and that their confidence intervals overlap. **Table 5** displays the estimates (with bounds of the 95% confidence interval) of the two dependence parameters $\chi$ and $\eta$ from the six models. These estimates converge toward a very weak asymptotic dependence. However, the estimation of dependence parameters in near independence is highly uncertain (**Sect. 3.3.2**).



**Table 5: Estimates of dependence parameters χ and η for extreme rainfall and wind gust at Heathrow airport for the time range 1971-2018**

| Models | Cond-Ex | JT-KDE | Gumcop | Normalcop | FGMcop | GalambosCop |
|--------|---------|--------|--------|-----------|--------|-------------|
| χ | 0.01 [0.00,0.02] | 0.06[0.05,0.09] | 0.04[0.01,0.06] | 0[0,0] | 0[0,0] | 0.04[02,0.06] |
| η | 0.49[0.45,0.54] | 0.54[0.49,0.59] | 1[1,1] | 0.52[0.51,0.54] | 0.5[0.5,0.5] | 1[1,1] |

### 4.2 Wildfire and extreme temperature in Portugal (Asymptotic dependence)

It has been shown that dry and warm conditions increase the risk of wildfire (AghaKouchak et al., 2018; Zscheischler et al.,
2018). Witte et al. (2011) established a direct link between a persistent heatwave and wildfire outbreaks in Russia and Eastern Europe in 2010.

The Northern Mediterranean countries (Portugal, Spain, France, Italy and Greece) are particularly affected by summer fires (Vitolo et al., 2019). Among these, Portugal holds the highest number of wildfires per land area (Pereira et al., 2011). There are many environmental and anthropogenic factors influencing the rural fire regime in Portugal and making its territory a fire
prone area. However, the majority of rural fires are recorded during hot and dry conditions in the summer (Pereira et al., 2011). Here, we used the mainland continental Portuguese Rural Fire Database, that includes 450,000 fires and covers the period 1980-2005 (Pereira et al., 2011), and includes data for all 18 districts in Portugal. This database is the largest such database in Europe in terms of total number of recorded fires in the 1980–2005 period (Pereira et al., 2011) and includes fires recorded down to a size of 0.001 ha. From the Portuguese Rural Fire database, we chose to focus on the Porto district, which was the
worst affected in the period (out of the 18 Portugal districts) in term of number of wildfires with 21.6% of the total fire recorded in the dataset between 1980 and 2005. The Porto district is situated in the northern part of Portugal (see **Fig. 11**), has an area of 2395 km² and is one of the most populated districts of Portugal with an estimated population of 1,778,146 in 2018 (Instituto Nacional de Estatística Portugal, 2019).

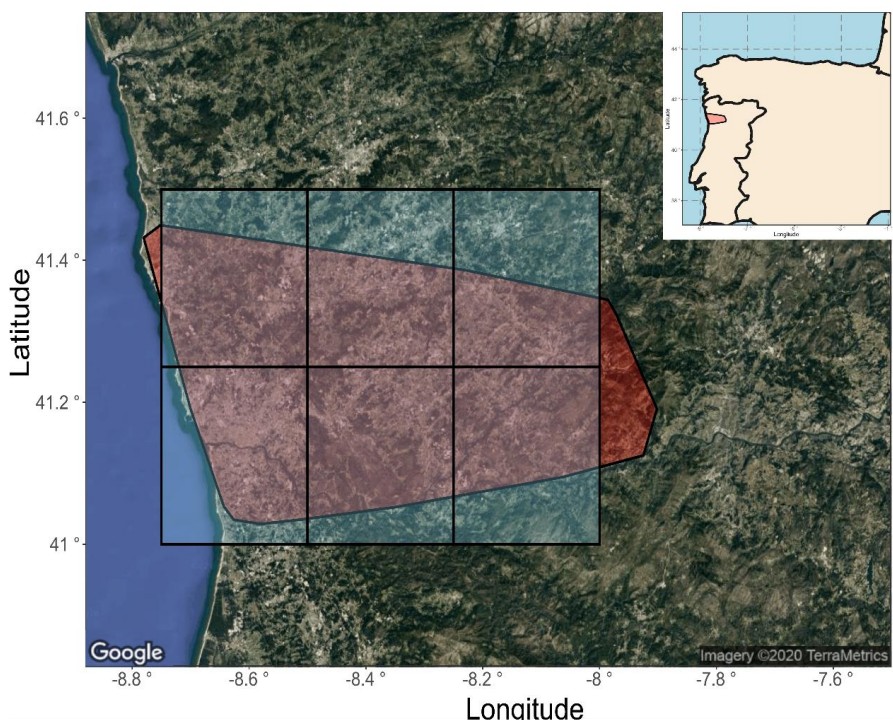

**Figure 11: Portugal study area for the interrelation between extreme hot temperature and wildfire burned areas. The red area represents the Porto district in Portugal containing studied wildfire burned areas. The blue tiles represent cells from the high-resolution gridded data set of daily climates over Europe (E-OBS) (Cornes et al., 2018) containing mean daily temperature data.** Satellite image retrieved with ggmap (Kahle and Wickham, 2013). © Google Maps (2020).

The bivariate dataset used to study the interrelation between extreme temperature and wildfire burned areas in the Porto district is composed of the following data:

a) *Daily number of wildfire (f).* Daily number of wildfires for the 26-year period 1980–2005 for the Porto district were extracted from the Portuguese Rural Fire Database dataset from Pereira et al. (2011). To account for under sampling of smaller wildfires in earlier years, and as suggested by Pereira et al. (2011), we used only those fires with a burned area $A_F \geq 0.1$ ha, resulting in 59,522 fires, an average of 6.3 fires per day (for those days with at least one fire occurrence) over the Porto district in Portugal (2395 km$^2$).

b) *Daily temperature data (t).* Daily mean temperature was extracted from the E-OBS gridded dataset (Cornes et al., 2018). We approximate the area in red in **Fig. 11** (Porto district) for each day with one temperature value by taking the average of daily temperatures in each of the six 0.25° x 0.25° cells represented by blue rectangles in **Fig. 11**. This assumption reduces the confidence in return level values and adds up with other interpolation uncertainties arising from the data (Hofstra et al., 2009). Moreover, the temperature in the six cells are strongly correlated (Pearson correlation coefficient $\rho > 0.98$) and variations in temperature are mostly due to the distance to the sea and altitude (Miranda et al., 2002).

The 26 years from 1980–2005 have a total of 9496 days. Of these, a total of 3442 days (36% of the days) have both non-zero days for number of wildfires and a mean temperature value, which are used in our final bivariate dataset. An overview of both daily mean temperature and daily number of wildfires is displayed in **Fig 12** in the form of monthly violin plots.


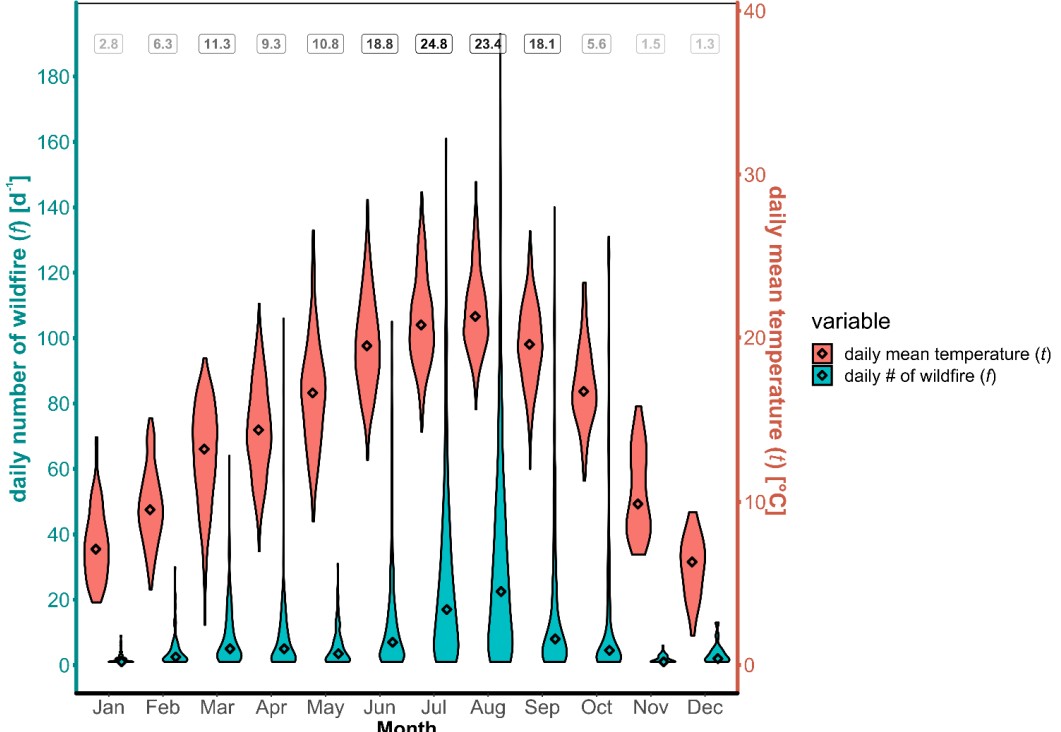

**Figure 12: Violin plot of those days with both daily mean temperature (red, upper violin plots) *t* and daily number of wildfires (blue, lower violin plots) *f* ≥ 1 fire d⁻¹, by month for the period 1980–2005 in Porto district (Portugal). Only those wildfires with burned area $A_F ≥ 0.1$ ha are included. Diamonds for both temperature and wildfires represent the median of all values in that month over the period of record. Numbers at the top of the graph represent the average number of days per month where there are recorded both a temperature value *t* and at least one wildfire (*f* ≥ 1 fire d⁻¹). Daily mean temperature data from E-OBS (Cornes et al., 2018) and wildfire data from Pereira et al. ( 2011).**


From **Fig. 12** we observe the seasonality in daily mean temperature with January the coldest month (median = 8.3°C) and

August the warmest (median = 21.0°C). Daily number of wildfires (with burned area $A_F ≥ 0.1$ ha) per month varies between median of 1.0–2.5 fire d⁻¹ in winter months (November to February) and 7.0–22.5 fire d⁻¹ in summer months (from June to September). The dataset is also represented as a scatterplot in **Fig. 13**. The scatterplot will be used for the model selection methodology presented at the beginning of **Section 4.**




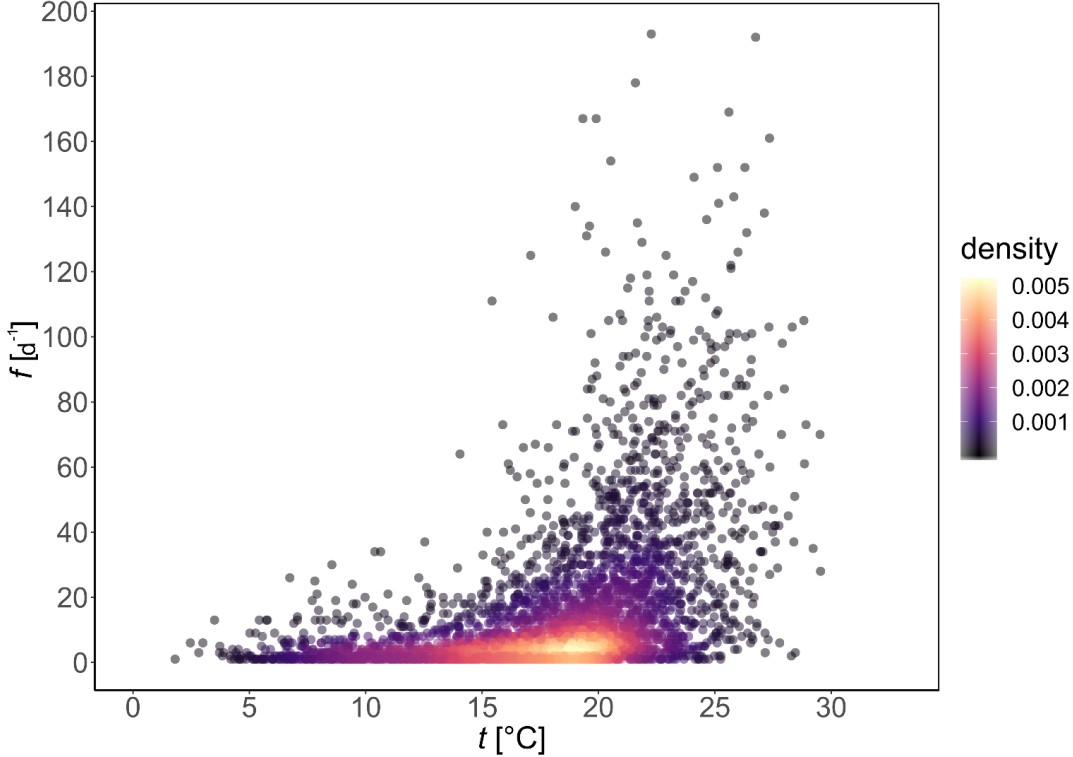


**Figure 13: Scatter plot of temperature as a dependence of wildfire occurrence in Porto district, Portugal for the period 1980–2005, for those days where there are recorded both mean daily temperature ($t$) and at least one fire, with $f$ the number of wildfires in one day. Only those wildfires with burned area $A_F \geq 0.1$ ha are included. Daily mean temperature data from E-OBS (Cornes et al., 2018) and wildfire data from Pereira et al. (2011).** Colours represent the bivariate density estimated from a kernel density estimator.

As discussed in the beginning of this section, extreme (hot) temperature and wildfire are interrelated. Indeed, extreme (hot) temperature may promote the development of wildfires (Witte et al., 2011; Sutanto et al., 2020) According to Tilloy et al. (2019), this is a change condition interrelation (i.e., one hazard changes environmental parameter that moves toward a change in the likelihood of another hazard). We then estimate the conditional exceedance probability curve (**Sect. 2.3**).

We now go through the four steps introduced at the beginning of **Sect. 4**.

(i) From **Figs. 13** and **6**, we can hypothesize that over our time range, the mean daily temperature and the number of wildfire per day are either asymptotically dependent ($\chi = 0.5$) or asymptotically independent but with strong positive association ($\eta = 0.9$) and that one marginal distributions have a slightly small shape parameter and the other one is heavily right-skewed (**AC**, **BC**).

(ii) This then gives us four analogous datasets and it is then possible to know from **Fig. 8** which models are the most
adapted to these conditions. The four datasets are the following:

- $\chi = 0.5$ and **AC**
- $\chi = 0.5$ and **BC**
- $\eta = 0.9$ and **AC**
- $\eta = 0.9$ and B-C

(iii) The confidence score for each model is the average of the *wd* from the four aforementioned datasets, and are presented in **Table 6**. Again, we have two asymptotically dependent and two asymptotically independent datasets. Based on the **Table 6**, the normal copula and FGM copula do not seem suitable to model the joint occurrence of wildfire and extreme temperature as these poorly fit the four datasets. The Gumbel and Galambos





copula have a confidence score of respectively 0.05 and 0.06 even though these are asymptotically dependent.
The joint tail-KDE model also has a confidence score of 0.05 while the conditional extremes model has a score

of 0.04.

According to these three first steps, we can identify the most relevant model for the bivariate dataset of daily maximum

temperature and daily wildfire occurrence in Porto district: the Gumbel copula, Galambos copula, the JT-KDE model and the

conditional extremes model are the most relevant models for our dataset.

**Table 6: Weighted Euclidian distance (*wd*) for datasets 1 to 4 based on extreme temperature-wildfire and six models, along with confidence scores (average of the *wd* for dataets 1 to 4). In bold are highlighted the four models with confidence scores < 0.1.**

| Dataset | Cond-Ex | JT-KDE | Gumbelcop | Normalcop | FGMcop | Galamboscop |
|---------|---------|--------|-----------|-----------|--------|-------------|
| 1 | 0.03 | 0.04 | 0.02 | 0.12 | 0.29 | 0.02 |
| 2 | 0.05 | 0.06 | 0.04 | 0.18 | 0.42 | 0.04 |
| 3 | 0.03 | 0.03 | 0.07 | 0.02 | 0.26 | 0.07 |
| 4 | 0.05 | 0.05 | 0.07 | 0.14 | 0.37 | 0.10 |
| Average | **0.04** | **0.05** | **0.05** | 0.12 | 0.34 | **0.06** |

(iv)    For illustration and/or confronting of the models with the data, the six models are fit to the dataset and the joint

exceedance level curves are produced with a joint exceedance probability set at p = 0.001, corresponding to a

bivariate return period of approximately 8 years.

In **Fig. 14** are displayed the conditional level curves produced from the four models that were selected after steps (i) to

(iii) and shown in bold values in **Table 6** (Cond-Ex, JT-KDE, Gumbelcop and GalambosCop).


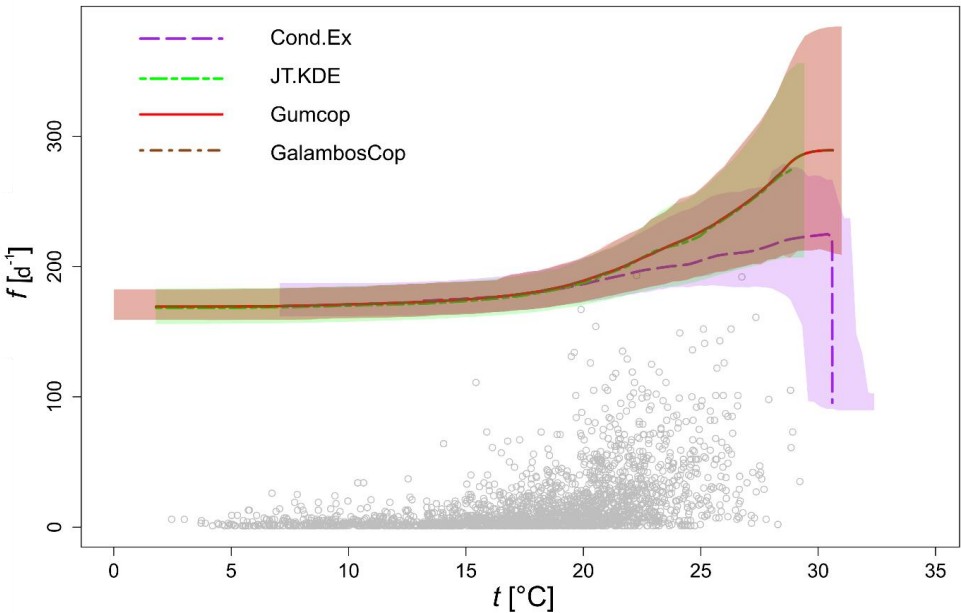

**Figure 14: Level curves for a *P*$_{and}$ joint probability *p*=0.001 of daily mean temperature and daily number wildfire occurrences in Porto district, Portugal, for the period 1980-2005. Level curves from the four models selected through the model selection methodology are displayed.**




From **Fig. 14,** we can observe that the JT-KDE and the Gumbel copula produce very similar conditional exceedance curves and that their confidence intervals highly overlap. However, the conditional extreme model provides a lower estimate than the other approaches the number of wildfire conditioning on the temperature being above a given threshold.

In **Table 7**, we present the estimates (with bounds of the 95% confidence interval) of the two dependence parameters $\chi$ and $\eta$ from the six models provide a bit more insight about the dependence structure. These estimates converge toward a moderate

asymptotic dependence varying from $\chi = 0.15$ (Cond-Ex) to $\chi = 0.47$ (GumCop). Even if all models tend to show asymptotic dependence between the two variables, estimates of $\eta$ are less than 1.0 for the normal copula, the JT-KDE model and the Cond-Ex with values varying between 0.67 and 0.79. This still implies a positive association between the two variables.

**Table 7: Estimates of dependence parameters χ and η for mean daily temperature and daily occurrences of wildfire in Porto district for the period 1980-2005**

| Models | Cond-Ex | JT-KDE | Gumcop | Normalcop | FGMcop | GalambosCop |
|---|---|---|---|---|---|---|
| $\chi$ | 0.15 [0.06, 0.20] | 0.26 [0.21, 0.30] | 0.47 [0.45, 0.49] | 0.00 [0.00, 0.00] | 0.00 [0.00,0.00] | 0.46 [0.44, 0.49] |
| $\eta$ | 0.67 [0.59, 0.72] | 0.67 [0.62, 0.71] | 1.00 [1.00, 1.00] | 0.79 [0.78,0.80] | 0.50 [0.50, 0.50] | 1.00 [1.00, 1.00] |

**5 Discussion and Conclusions**

Quantifying and measuring the interrelations between different natural hazards is a key element when adopting a multi-hazard approach (Gill and Malamud, 2014; Leonard et al., 2014). In this study, we focused on statistical approaches that are often used to characterize and model interrelations between hazards. Another focus has been on modelling relationships between hazards at an extreme level. In total six statistical models with different characteristics (nature of asymptotic dependence,

parametric/semi-parametric) were compared. Some of these models have already been used to study compound extremes in hydrology and climatology (Hao et al., 2018; Liu et al., 2018; Sadegh et al., 2018; Cooley et al., 2019). However, these have not been compared over a broad range of bivariate datasets and applied to the same natural hazards in the same location.

This section will discuss the four following themes before a short conclusion: (a) choices influencing the results of the simulation study; (b) uncertainties at the interface between asymptotic dependence and asymptotic independence; (c) possible

extensions of this approach to more than two hazards.

*Choices influencing the results of the simulation study*. This study aimed to assess the fitting ability of several bivariate models to a broad range of datasets. In order to do so, models were compared on their ability to reproduce an extreme level curve (see **Sect. 3.2.1**). The extreme level curve probability was set at $p = 0.001$. The multivariate regular variation framework (Resnick, 1987) provides evidence supporting the fact that the dependence structure remains identical in the whole extreme domain.

However, some results shown in **Sect. 3.3** might have been influenced by the value of the joint exceedance probability. In particular, it is likely that when decreasing the level curve probability (i.e., to more extreme values), the flexibility and abilities of the asymptotically independent normal copula will decrease. There are many copulas other than the four selected in this study (Nelsen, 2006; Sadegh et al., 2017) that have been developed. Nevertheless, we believe the four copulas used in this study are suitable for bivariate extreme value analysis and are amongst the most widely used in the literature (Genest and

Favre, 2007; Genest and Nešlehová, 2013). Another influential choice in this study has been the number of synthetic data points generated in each realization of the dataset. The number of data points and data set size is an important influence on uncertainty in natural hazard modelling and probabilistic approaches (Frau et al., 2017; Liu et al., 2018). For each simulation, we simulated $n = 5000$ data points. Some other simulation studies took a higher number of data points (Zheng et al., 2014; Cooley et al., 2019); however, by we replicated 100 times and produced confidence intervals, thus ensuring consistency of our

results. We also found that threshold selections, to fit the generalised Pareto distributions of the marginal distributions and to estimate the extremal dependence measures, also have an influence on our results.

*Uncertainties at the interface between asymptotic dependence and asymptotic independence.* From the results of the simulation study (**Sect. 3.3**) and the two case study applications (**Sect. 4**), one can observe that the interface between asymptotic





dependence and asymptotic independence can be unclear. In **Sect. 3.3**, we discussed the decrease in model performance and
the increase in uncertainty for low values of $\chi$ and high values of $\eta$. Taking the assumption of asymptotic independence or
asymptotic dependence can have a significant impact on the estimation of joint return levels. We find that extra care is required
when dealing with bivariate datasets which are near independence as in **Sect. 4.1**.

*Possible extension of the approaches to more than two hazards*. As presented through this paper, the study of interrelations
between natural hazards has primarily been done by hazard pairs (e.g., Gill and Malamud, 2014). Dependence measures and a
variety of different models or level curves, all presented in this article, are powerful tools to assess, quantify and model
interrelations between two hazards. However, in many cases, multi-hazard events include more than two hazards interacting
in various ways (e.g., Gill and Malamud, 2014; Leonard et al., 2014). The use of models presented in this article can be
extended to more than two variables, sometimes with disadvantages. One of these disadvantages is the that the parametric
nature of copulas lead to a lack of flexibility while going to higher dimensionality (Bevacqua et al., 2017; Hao et al., 2018).
The JT-KDE and Cond-Ex models are suitable for higher dimensions (Davison and Huser, 2015; Cooley et al., 2019), although,
these have not been tested for high dimensional multi-hazard modelling yet (Tilloy et al., 2019). Recent research conducted
suggest pair-copula construction (PCC, also known as vine copula) (Bedford and Cooke, 2002; Hashemi *et al.*, 2016; Bevacqua
*et al.*, 2017, Lui *et al.*, 2018) and non-parametric Bayesian networks (NPBN) (Hanea et al., 2015; Couasnon et al., 2018) can
be used to model multi-hazard events with more than two hazards. The vine copula framework allows one to select different
bivariate copulas for each pairs of variables, providing a great flexibility in dependence modelling (Brechmann and
Schepsmeier, 2013; Hao and Singh, 2016). Non-parametric Bayesian networks, which are associated with the structure of
Bayesian network and copulas (Hanea, 2010; Hanea *et al.*, 2010, 2015), have been used to study multiple dependences between
river discharge and storm surges in the USA during a hurricane (Couasnon *et al.*, 2018).

In conclusion, we have compared and examined the strength and weaknesses of six distinct bivariate extreme models in the
context of hazard interrelations. These six models are grounded in multivariate extreme value theory and represent the diversity
of approaches (e.g., non-parametric vs parametric) currently applied to hazard interrelation analysis. With this study we aimed
to contribute to a better understanding on the applicability of bivariate extreme models to a wide range of natural hazard
interrelations. The methodology developed in this article is aimed to be widely applicable to a variety of different hazards and
different interrelations, here represented by the 60 synthetic datasets created. Abilities of each model have been assessed with
two metrics: (i) dependence measure; (ii) bivariate return level (level curves). These two metrics and the different diagnostic
tools developed in this study offer new intuitive ways to decipher the dependence between two variables. We recommend
selecting a range of models rather than one when studying interrelations between two hazards. To highlight the benefits of the
systematic framework developed, we studied the dependence between extremes (natural hazards) of the following
environmental data: (i) daily precipitation accumulation and daily maximum wind gust (maximum over a period of 3 s) at
Heathrow airport (UK) over the period 1971-2018; (ii) daily mean temperature and daily number of wildfires in Porto district,
Portugal over the period 1980-2005. The two datasets represent different hazard interrelations: (i) compound interrelation
between extreme wind and extreme rainfall and (ii) change condition interrelation where higher air temperature change
condition for wildfire occurrence. In both cases, a sample of the most relevant model among the six used in this study have
been selected and fitted to the bivariate datasets. The good agreement in the estimation of bivariate return period between
models corroborate the relevance of the comparison metrics we developed.





## Appendix A: Comparing model abilities through tail dependence measures

### A.1. Tail dependence measures estimations

Tail dependence measures $\eta$ and $\chi$ are estimated by each model. For copulas, these measures are related to the copula
parameters. In our set of four copulas, two are asymptotically dependent (Gumbel and Galambos) with $\eta=1$ and two are
asymptotically independent (normal and FGM) with $\chi=0$.

For the nonparametric joint tail approach, the $\chi$ and $\eta$ measures are estimated following the procedure used by Winter (2016).
For the conditional model, both measures are estimated from the simulated points. Marginal distributions $(X_1, X_2)$ are
transformed to the uniform margins (U1, U2). The $\chi$ measure is estimated by calculating the probability $P(V > u | U > u)$ (**Eq.**
**4**). The $\eta$ measure is estimated in two steps. First we estimate $\bar{\chi}(u)$ as (Coles et al., 1999):

$$\bar{\chi}(u) = \frac{2\log P(U>u)}{\log P(U>u,V>u)} - 1 = \frac{2\log(1-u)}{\log(\chi(u)(1-u))} - 1, \tag{A1}$$

for $0 \leq u \leq 1$ with $u$ a sufficiently high threshold. Second, the $\eta$ measure is estimated from $\bar{\chi}$ (**Eq. A1**).

To compare the estimated dependence measure to the reference value, the root-mean-square error (RMSE), a measure of
efficiency that accounts for both the bias and variance of the estimates is used, similarly to Zheng et al. (2014).

### A.2. Comparison of model abilities

The estimation of dependence measure is an important step in bivariate analysis (Coles et al., 1999; Heffernan, 2000; Zheng
et al., 2013, 2014; Dutfoy et al., 2014). Models have also been compared on their ability to estimate the dependence measures
$\chi$ and $\eta$ Results arising from this comparison provide a different perspective on the abilities of each models. **Figure A1**
shows the RMSE of the dependence measures estimations for each of the 60 synthetic datasets.





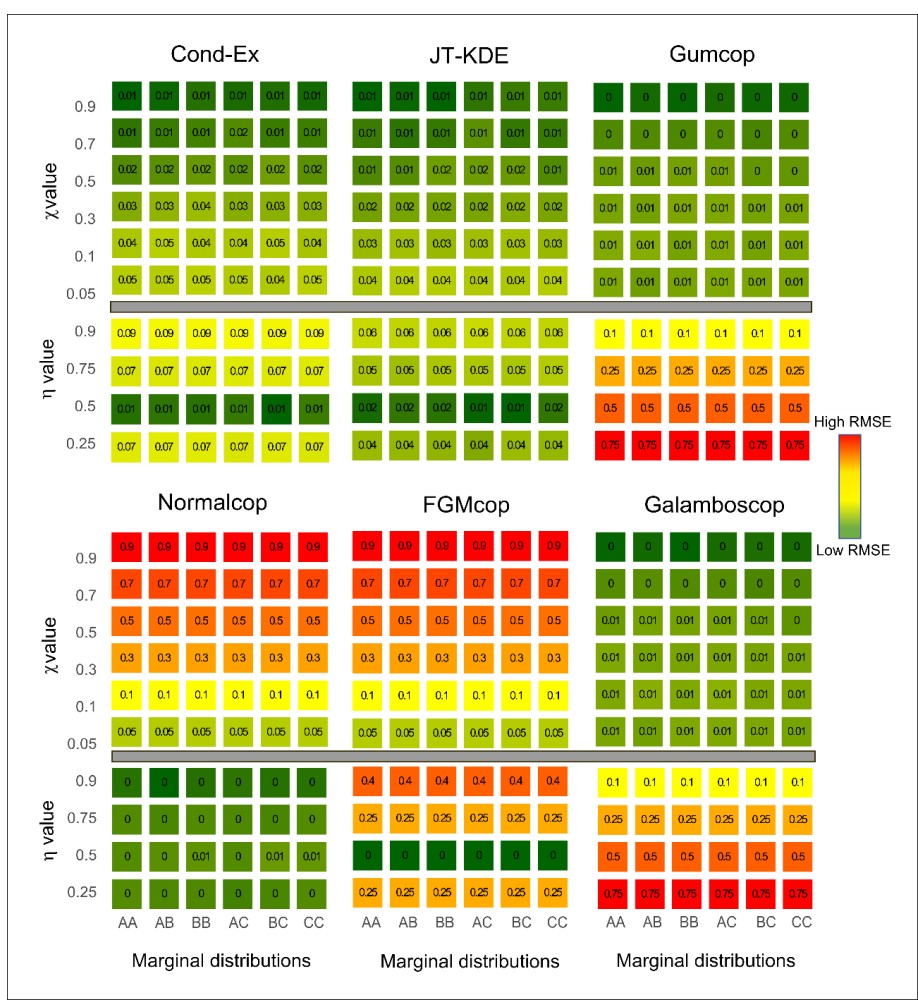

**Figure A1: RMSE (root-mean-square error) of the estimated dependence measures to the reference for all 60 different datasets. Fitting capacities of each model are represented. Values in cells and colours represent the median RMSE from low (dark green) to high (red). Thickness of cell borders represent the 95% uncertainty around the median value**

From **Fig. A1**, we observe the following:

- Marginal distributions do not have a significant impact on the accuracy of the estimation of these measures for the copulas.

- Marginal distributions have a small impact on estimation of the dependence measures for the conditional extremes model and the joint-tail model, however this impact is not as important as for the level curve estimation

- All copulas estimate very accurately the dependence measure within their operating range (AI for normal copula, near independence for FGM copula and AD for Gumbel and Galambos copula). However, only the conditional extremes model and the joint-tail model can estimate both $\eta$ and $\chi$.

- The dependence measure estimator used in the joint-tail KDE approach offers slightly more accurate estimation for, in particular for $\eta$.

- Estimation performance of both joint tail KDE and condition extreme models decrease when approaching the interface between asymptotic dependence and asymptotic independence. The RMSE at $\chi=0.05$ is close the 100% of





the value of $\chi$ while the RMSE $\eta = 0.9$ is at its highest for both Cond-EX and JT-KDE models. It is then hard to decipher with confidence the nature of the dependence in the asymptotic domain for low $\chi$ values and high $\eta$ values.



**Code availability**

The codes used to produce synthetic data and the 6 models used in this study are publicly available on GitLab: https://gitlab.com/doudeg/bivariate_models.git

**Data availability**

We acknowledge the E-OBS dataset from the EU-FP6 project UERRA (http://www.uerra.eu) and the data providers in the
ECA&D project (https://www.ecad.eu). The wind gust data was obtained from the UK Met Office. The fire data is available from Pereira et al. (2011). The bivariate (wildfire, temperature) data used is available at: https://gitlab.com/doudeg/bivariate_models.git.

**Author contribution**

All authors discussed the whole plan of this article, Tilloy designed and implemented all the experiments, prepared all the data
and finish the draft, including all figures in the article. Malamud, Winter and Joly-Laugel revised the article. Winter supported the methods and techniques.

**Competing interests** Malamud is on the editorial board of this journal (the article has been overseen independently by another editor of this journal).


**Special issue statement.** This research article is part of the special issue "Advances in extreme value analysis and application to natural hazards".

**Acknowledgment**

Funding: The first author was supported by an EDF R&D PhD studentship.

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
