# Peer review of "Evaluating the efficacy of bivariate extreme modelling approaches for multi-hazard scenarios"

_Natural Hazards and Earth System Sciences, 2020_

## Referee Comment (RC1) · Anonymous Referee #1 · 16 Mar 2020

General Comments:

The paper assesses the capability of six bivariate statistical approaches to model 60 distinct synthetic (bivariate) datasets, some of which possess asymptotic independence and others asymptotic independence. The results are used to develop a systematic framework for selecting among the competing statistical models. The framework is then demonstrated by way of two real world examples. The framework offers a novel approach for selecting among multivariate models. The manuscript is generally well written and relevant to the topic of "Advances in extreme value analysis and application to natural hazards". In the opinion of this reviewer it is therefore worthy of publication in this special issue of Natural Hazards and Earth System Science. Nevertheless, the manuscript would benefit from a slight reorganization, additional explanation at certain

points and a thorough review of grammar and spelling.

Specific comments:

The abstract is missing a sentence describing the link between the synthetic datasets part and the application of the framework to environmental data (i.e., the work on the synthetic datasets aids in the creation of the framework). At present, the abstract states that the benefit of a systematic modeling framework are highlighted without any introduction/description of the framework.

The manuscript would benefit from a more precise definition of what constitutes a hazard in relation to other recent literature on compound and cascading hazards. For instance, Zscheischler et al. (2018) defines compound events as "The combination of multiple drivers and/or hazards that contributes to societal or environmental impacts.". Therefore, they may consider rain, lightening and hail as drivers and a landslide as a hazard, whereas here all four are considered as hazards.

P2 L49 to P3 L83 in the introduction focuses on methodology and introduces specific subsections of the methodology section before the methodology section is introduced in the final (roadmap) paragraph at the end of the introduction. I recommend moving the text to the start of the methodology section. Perhaps, a paragraph giving a broad summary of the synthetic dataset work and the modelling framework including the link between them could be added.

For the wildfire example, would $\chi$=0.7 with marginals AC also be a relevant test case.

The (subjective) selection of the AND should also be discussed in the "Choices influencing the results of the simulation study" part of the conclusion.

Technical corrections:

Abstract: "match" implies that synthetic data was based on specific examples of environment data "be representative of" maybe a more appropriate expression.

P2 L46: "Copulas" refers to a specific type of model whereas "multivariate model" is more general, perhaps replace the latter with a more specific description of the models.

P4 L94-95: The sentence "A theoretical background on extreme value theory is given in Supplement S1.1." should appear earlier in the paragraph, after the sentence which starts "Extreme Value Theory . . .". P4 L106: Typo. Remove "Then".

P4 L116: Typo. Remove ".".

P6 L166: Typo. Replace "F_X2 (x_2))" with "F_X2 (x_2)".

P6 L170: Use "e.g.," here and elsewhere before citations where there are other relevant papers omitted due to the need for brevity.

P7 L175: "Formally, the application of a copula model can be summarized in four main steps". It should be made clear here and elsewhere that "the application" refers to the application in this study and may vary elsewhere for instance in terms of marginal distributions etc.

P7 L181: "joint distribution" may refer to the full multivariate distribution including marginals. Consider changing to "dependence structure" or similar.

P7 L210: "extrapolate the conditional model to simulate new extreme data." Do you do this?

P8 L218: Subjective term. Remove "very".

P11 L186: Perhaps change "amount" to "number" to be more specific.

P13 L323: "Confront" seems like an unusual term to use here. Consider replacing it with "compare and contrast" or simply remove "confront or".

Table 2: Spelling. Change "thorough" to "throughout". P13 L334: Typo. "Random variables x". Consider capitalizing X and Y as they denote random variables.

P14 L338: Why characterize log-normal distributors by the coefficient of variation?

[Figure]

Table 3: Typo. Replace "dataset" with "datasets".

P16 L379 Typo. Add space "test(". P16 L378 & L379 & L392: Typo. Remove first names "(Arnold, Taylor and Emerson, John, 2011)".

P16 L389: Grammar. "issue is" or "issues are".

P16 L391-393: "The measures mentioned above are not suitable as they imply parametric distributions to be compared against observations (Stephens, 1970; Arnold, Taylor and Emerson, John, 2011). It is then not possible to compare the goodness-of-fit on the whole range of the data." I am not clear as to the exact limitation being discussed here.

P16 L399: A description of the reference level curve is required that it comes from the "underlying bivariate (X_1,X_2) distribution of the data" as stated in Figure 6 should be added to the main body of text.

P19 L461: Consider adding "in general" or similar before "outperform all the other...".

P19 L468-469: Grammar. Rephrase "Gumbel and Galambos copulas show very similar behaviours with respectively 68% [53–93%] and 68% [52–93%] cases with wd < 0.1.".

P20 L473 Grammar. Change "less" to "least".

P20 L494 I believe "Fig. 6" should be "Fig. 5".

P20 L497 Grammar. Change "abilities of each models for the selected datasets in step (ii)" to "abilities of each model for the datasets selected in step (ii)".

P20 L510 & P21 L519: "London Heathrow airport, UK" and "Heathrow airport (London UK)" be consistent with names, only need to specify it is in the UK on first mention.

P21 L515: Typo. Add spaces: "Season(Hawkes".

P22 L544-545: Repetition: Remove "Daily rainfall data from E-OBS (Cornes et al.,

2018) and wind gust data (maximum 3 s wind 545 velocity in a day) from the Met Office (2019)." as the information is already given in the text.

P23 L555-556: Repetition. Remove "Daily rainfall data from E-OBS (Cornes et al., 2018) and wind gust data (the 555 maximum 3 s wind velocity in a day) from the Met Office (2019)." as the information is already given in the text.

P23 L560: I believe "Figs. 9 and 6" should read "Figs. 9 and 5".

P23 L565 - 568: Consider replacing circle bullet points with numbers 1-4 so the cases correspond to the relevant rows of Table 4.

P23 L573: Add "average" before "confidence score".

P24 L578: Grammar. Replace "model" with "models".

P24 L591: Grammar. Replace "curve" with "curves".

P25 L612 & P25 L612: "2395 km2" and "1,1778,146" be consistent with use of commas to separate numbers here and throughout the manuscript.

P27 L643 – 644: Repetition. Remove "Daily mean temperature data from E-OBS (Cornes et al., 2018) and wildfire data from Pereira et al. (2011)." as the information is already given in the text.

P28 L653-654: Repetition. Remove "Daily mean temperature data from E-OBS (Cornes et al., 2018) and wildfire data from Pereira et al. (2011). as the information is already given in the text.

P28 L660: I believe "Figs. 13 and 6" should read "Figs. 13 and 5".

P28 L666 - 669: Consider replacing circle bullet points with numbers 1-4 so the cases correspond to the relevant rows of Table 6.

P28 L666 - 669: Replace "B-C" with "BC".

P30 L713: Typo. Replace "four following" with "following three".

P31 L742-744: Grammar: Change "One of these disadvantages is the that the parametric nature of copulas lead to a lack of flexibility while going to higher dimensionality." to "One of these disadvantages is that the parametric nature of copula leads to a lack of flexibility when going to higher dimensionality." or similar.

P31 L747: Pair Copula Construction and vine copula are not synonymous rather vine copulas are a type of Pair Copula Construction.

P32 L787: Typo. "$\eta$Results" add "."

P32 L783: Perhaps add a sentence to make it clear that the RMSE comes from the calculation of the dependence measures for the 100 realizations of the 60 datasets.

P36 L887: Remove "pp."

References:

Zscheischler, J., Westra, S., Van Den Hurk, B.J., Seneviratne, S.I., Ward, P.J., Pitman, A., AghaKouchak, A., Bresch, D.N., Leonard, M., Wahl, T. and Zhang, X., 2018. Future climate risk from compound events. Nature Climate Change, 8(6), pp.469-477.

---

## Referee Comment (RC2) · Anonymous Referee #2 · 23 Mar 2020

This study evaluates the efficacy of various bivariate models for a variety of correlation structures in a large set of synthetic data. While I am not a fan of synthetic data, the rigorous and comprehensive evaluation of various models justify the publication of this paper in NHESS. However, there are some issues that remain to be resolved before final publication:

1. There is a large focus on multi-hazard analysis in this paper. However, I cannot see the sector that is impacted by the presented case studies. For a multi-hazard coastal flood analysis, for example, it is intuitive how higher water levels and larger river flows lead to a larger risk of flooding. However, who is impacted by rainfall and wind? What is the final impact that is worsen by concurrence and sub-sequence of multiple hazards? Even more confusing is the wildfire case study. Wildfire is the impact

of high-temperature, hence I am not clear whether it is justified to consider the wildfire and temperature as two drivers of one impact. What is going to be that impact? How is it worsen by the combination of these two hazards? The manuscript should justify the presented multi-hazard analyses.

2. Wildfire burned area is related to temperature not the number of fires. A fire of size 0.1 ha can occur all year around, specially for human started fires (see Balch, J. K., et al. (2017). Human-started wildfires expand the fire niche across the United States. Proceedings of the National Academy of Sciences, 114(11), 2946-2951.). But more importantly, what is the impact? If it is the wildfire, is it justified to have fire as the impact and as the hazard? Is high-temperature necessarily a hazard?

3. Abstract needs some improvement. I struggled to understand how the case studies are related to synthetic data.

4. An important missing element in the evaluation of models is p-value. It would be interesting to see what the p-values are and determine whether the models fail/pass to represent the data! This is actually very important. The metrics used in the paper are subjective - although valuable - and a more widely accepted metric could help the general audience relate the study to other modeling practices.

5. There are many typos in the text. I highlighted some of them in the attached manuscript, but there are more.

6. There are some specific comments that are provided in the attached manuscript.

Hope the authors find the comments useful to improve their paper

Please also note the supplement to this comment:
https://www.nat-hazards-earth-syst-sci-discuss.net/nhess-2020-28/nhess-2020-28-RC2-supplement.pdf
* * *
2020-28, 2020.

[Figure]

**Supplement:**

[revised manuscript text omitted]

---

## Author Comment (AC1) · 10 Jun 2020

[R1 Introduction] The paper assesses the capability of six bivariate statistical approaches to model 60 distinct synthetic (bivariate) datasets, some of which possess asymptotic independence and others asymptotic independence. The results are used to develop a systematic framework for selecting among the competing statistical models. The framework is then demonstrated by way of two real world examples. The framework offers a novel approach for selecting among multivariate models. The manuscript is generally well written and relevant to the topic of "Advances in extreme value analysis and application to natural hazards". In the opinion of this reviewer it is therefore worthy of publication in this special issue of Natural Hazards and Earth System Science. Nevertheless, the manuscript would benefit from a slight reorganization,

additional explanation at certain points and a thorough review of grammar and spelling.

[AR to General Comments] We thank the Anonymous Reviewer 1 (R1) for this summary, and suggestion of reorganization, additional explanation and thorough review of grammar and spelling. Based on these comments, we have revised the manuscript, particularly restructuring parts of our manuscript. Below, we provide detailed replies to the reviewer 1 comments (Author Replies—AR).

[R1 Specific comments]

[R1 Comment 1] The abstract is missing a sentence describing the link between the synthetic datasets part and the application of the framework to environmental data (i.e., the work on the synthetic datasets aids in the creation of the framework). At present, the abstract states that the benefit of a systematic modeling framework are highlighted without any introduction/description of the framework.

[AR 1] We agree with the reviewer about the fact that a linking sentence is missing in the abstract. We therefore modified the abstract to define with more clarity the framework and its purpose (L10-11) and (L12-13). [R1 Comment 2] The manuscript would benefit from a more precise definition of what constitutes a hazard in relation to other recent literature on compound and cascading hazards. For instance, Zscheischler et al. (2018) defines compound events as "The combination of multiple drivers and/or hazards that contributes to societal or environmental impacts.". Therefore, they may consider rain, lightening and hail as drivers and a landslide as a hazard, whereas here all four are considered as hazards. [AR 2] We thank the reviewer for this comment as we agree on the importance of defining what constitutes a hazard. We recognize that there are different definitions. Here, the term hazard will follow the definition by UNISDR (2009), which refers to a natural hazard (hereafter referred to as a 'hazard') as a natural process or phenomenon that may have negative impacts on society. As hail, lightning and extreme rainfall fall in the category of "hazards". A sentence has therefore been added to the introduction.

[R1 Comment 3] P2 L49 to P3 L83 in the introduction focuses on methodology and introduces specific subsections of the methodology section before the methodology section is introduced in the final (roadmap) paragraph at the end of the introduction. I recommend moving the text to the start of the methodology section. Perhaps, a paragraph giving a broad summary of the synthetic dataset work and the modelling framework including the link between them could be added.

[AR 3] We agree with the reviewer on this. Parts of the mentioned paragraph has therefore been moved. Paragraph P2 L52-59 has been moved to the start of the methodology section (P4 L101-111). The sentences P3 L 80-83 is now incorporated in the "roadmap" paragraph (P3 L88-190).

[R1 Comment 4] For the wildfire example, would $\chi$=0.7 with marginals AC also be a relevant test case.

[AR 4] Thank you for this comment. It also raises the question of the selection of analogous datasets, which is currently quite subjective. To tackle this issue, we decided to add a small tool to help the selection. The tool is based on the empirical estimates of the 2 dependence measures ($\chi$ and $\eta$) and their uncertainty bounds. It therefore suggests analogous which have a dependence measure within the uncertainty bound. We believe this tool strengthen the whole methodology. Using this tool, $\chi$=0.7 with AC marginal is not a relevant test case. We also reviewed our previous choices to be in accordance with the selection tool: we added the analogous AB and AC with $\chi$=0.3 for wildfires and $\chi$=0.1 AB and BB.

[R1 Comment 5] The (subjective) selection of the AND should also be discussed in the "Choices influencing the results of the simulation study" part of the conclusion.

[AR 5] Very good point. We now state the potential influence of the selection of the "AND" probability in the conclusion (P32 L780-782).

[Reviewer 1 Technical corrections] [AR to R1 Technical corrections] We thank the Reviewer 1 for these detailed technical corrections. We have below replied to each one and taken on board the vast majority of them.

Abstract: "match" implies that synthetic data was based on specific examples of environment data "be representative of" maybe a more appropriate expression. We totally agree, thanks, we have now made the change. P2 L46: "Copulas" refers to a specific type of model whereas "multivariate model" is more general, perhaps replace the latter with a more specific description of the models. We replace "multivariate models" by "Multivariate extreme models". We did not find a more specific appellation.

P4 L94-95: The sentence "A theoretical background on extreme value theory is given in Supplement S1.1." should appear earlier in the paragraph, after the sentence which starts "Extreme Value Theory : : :". Change done

P4 L106: Typo. Remove "Then". Change done

P4 L116: Typo. Remove ".". Change done

P6 L166: Typo. Replace "$F\_X2 (x\_2))$" with "$F\_X2 (x\_2)$". Change done

P6 L170: Use "e.g.," here and elsewhere before citations where there are other relevant papers omitted due to the need for brevity. Change done.

P7 L175: "Formally, the application of a copula model can be summarized in four main steps". It should be made clear here and elsewhere that "the application" refers to the application in this study and may vary elsewhere for instance in terms of marginal distributions etc. Change done

P7 L181: "joint distribution" may refer to the full multivariate distribution including marginals. Consider changing to "dependence structure" or similar. Change done

P7 L210: "extrapolate the conditional model to simulate new extreme data." Do you do this? Not exactly, that sentence was misleading and not well written, it has been rephrased to improve clarity.

P8 L218: Subjective term. Remove "very". Change done

P11 L186: Perhaps change "amount" to "number" to be more specific. Change done

P13 L323: "Confront" seems like an unusual term to use here. Consider replacing it with "compare and contrast" or simply remove "confront or". Change done

Table 2: Spelling. Change "thorough" to "throughout". P13 L334: Typo. "Random variables x". Consider capitalizing X and Y as they denote random variables. Change done

P14 L338: Why characterize log-normal distributors by the coefficient of variation? This has been the topic of a long debate between authors. The idea behind using the coefficient of variation is to (i) relate to previous literature working with log-normal distribution. (ii) characterise the log normal distributions used with one single parameter instead of two. We clarified this by adding a sentence composed of the two aforementioned points (P14 L370-371). Table 3: Typo. Replace "dataset" with "datasets". Change done

P16 L379 Typo. Add space "test(". Change done

P16 L378 & L379 & L392: Typo. Remove first names "(Arnold, Taylor and Emerson, John, 2011)". Change done

P16 L389: Grammar. "issue is" or "issues are". Change done

P16 L391-393: "The measures mentioned above are not suitable as they imply parametric distributions to be compared against observations (Stephens, 1970; Arnold, Taylor and Emerson, John, 2011). It is then not possible to compare the goodness-of-fit on the whole range of the data." I am not clear as to the exact limitation being discussed here. The main limitation being discussed is the need for models to be parametric (e.g., copula) to be compared with the discussed method. As we worked in this study with parametric and non(semi) parametric models, it is not possible to use the previously discussed Goodness of Fit methods. We changed the text to make it clearer.

P16 L399: A description of the reference level curve is required that it comes from the "underlying bivariate ($X_1,X_2$) distribution of the data" as stated in Figure 6 should be added to the main body of text. Thank you, Change done

P19 L461: Consider adding "in general" or similar before "outperform all the other: : :". Change done

P19 L468-469: Grammar. Rephrase "Gumbel and Galambos copulas show very similar behaviours with respectively 68% [53–93%] and 68% [52–93%] cases with wd < 0.1.". Change done

P20 L473 Grammar. Change "less" to "least". Change done

P20 L494 I believe "Fig. 6" should be "Fig. 5". Change done

P20 L497 Grammar. Change "abilities of each models for the selected datasets in step (ii)" to "abilities of each model for the datasets selected in step (ii)". Change done

P20 L510 & P21 L519: "London Heathrow airport, UK" and "Heathrow airport (LondonUK)" be consistent with names, only need to specify it is in the UK on first mention. Change done

P21 L515: Typo. Add spaces: "Season(Hawkes". Change done

P22 L544-545: Repetition: Remove "Daily rainfall data from E-OBS (Cornes et al2018) and wind gust data (maximum 3 s wind 545 velocity in a day) from the Met Office (2019)." as the information is already given in the text. Change done

P23 L555-556: Repetition. Remove "Daily rainfall data from E-OBS (Cornes et al., 2018) and wind gust data (the 555 maximum 3 s wind velocity in a day) from the Met Office (2019)." as the information is already given in the text. Change done

P23 L560: I believe "Figs. 9 and 6" should read "Figs. 9 and 5". Change done

P23 L565 - 568: Consider replacing circle bullet points with numbers 1-4 so the cases

correspond to the relevant rows of Table 4. Good idea! Change done

P23 L573: Add "average" before "confidence score". We are unclear here if it makes sense to describe the "confidence score" as an average and would require further clarification from the reviewer.

P24 L578: Grammar. Replace "model" with "models". Change done

P24 L591: Grammar. Replace "curve" with "curves". Change done

P25 L612 & P25 L612: "2395 km2" and "1,1778,146" be consistent with use of commas to separate numbers here and throughout the manuscript. Change done, Thank you

P27 L643 – 644: Repetition. Remove "Daily mean temperature data from E-OBS(Cornes et al., 2018) and wildfire data from Pereira et al. (2011)." as the information is already given in the text. Change done

P28 L653-654: Repetition. Remove "Daily mean temperature data from E-OBS (Cornes et al., 2018) and wildfire data from Pereira et al. (2011). as the information is already given in the text. Change done

P28 L660: I believe "Figs. 13 and 6" should read "Figs. 13 and 5". Change done

P28 L666 - 669: Consider replacing circle bullet points with numbers 1-4 so the cases correspond to the relevant rows of Table 6. Change done

P28 L666 - 669: Replace "B-C" with "BC". Change done

P30 L713: Typo. Replace "four following" with "following three". Change done

P31 L742-744: Grammar: Change "One of these disadvantages is the that the parametric nature of copulas lead to a lack of flexibility while going to higher dimensionality." to "One of these disadvantages is that the parametric nature of copula leads to a lack of flexibility when going to higher dimensionality." or similar. Change done

P31 L747: Pair Copula Construction and vine copula are not synonymous rather vine

copulas are a type of Pair Copula Construction. Thank you for this comment, we changed the sentence accordingly.

P32 L787: Typo. "_Results" add "." Change done

P32 L783: Perhaps add a sentence to make it clear that the RMSE comes from the calculation of the dependence measures for the 100 realizations of the 60 datasets. Change done

P36 L887: Remove "pp." Change done
* * *
[Figure]

**Fig. 1.** Weighted Normalized Euclidean Distance (wd) to the reference curve for all 60 different synthetic datasets. Fitting capacities of each model are represented. Values in cells and colours represent the

---

## Author Comment (AC2) · 10 Jun 2020

[R2 Introduction] This study evaluates the efficacy of various bivariate models for a variety of correlation structures in a large set of synthetic data. While I am not a fan of synthetic data, the rigorous and comprehensive evaluation of various models justify the publication of this paper in NHESS. However, there are some issues that remain to be resolved before final publication:

[AR to General Comments] We thank the Anonymous Reviewer 2 (R1) for this summary. Below, we provide detailed replies to the Reviewer comments (Author Replies—AR).

[R2 Comment 1] 1. There is a large focus on multi-hazard analysis in this paper.

[Figure]

However, I cannot see the sector that is impacted by the presented case studies. For a multi-hazard coastal flood analysis, for example, it is intuitive how higher water levels and larger river flows lead to a larger risk of flooding. However, who is impacted by rainfall and wind? What is the final impact that is worsen by concurrence and subsequence of multiple hazards? Even more confusing is the wildfire case study. Wildfire is the impact of high-temperature, hence I am not clear whether it is justified to consider the wildfire and temperature as two drivers of one impact. What is going to be that impact? How is it worsen by the combination of these two hazards? The manuscript should justify the presented multi-hazard analyses.

[AR 1] Thank you for these thoughtful comments, which we respond to here including how we have modified our manuscript in response.

• To put this into context, two authors of the manuscript work for an energy company and the original framework of carrying out this research was within problems that might be of interest for the energy sector. We therefore better highlight the potential impact to society and energy infrastructure in our revised manuscript (P3 L76-84). • We agree with the reviewer that while compound flooding is measurable in water height (and thus the potential impacts this might have on various types of infrastructure), the potential impact of compound extreme wind and extreme rainfall is harder to quantify. In the context of energy infrastructure, the combination of extreme rainfall and extreme wind can lead, for example, to power cuts, wind destroying roof leading to greater damages, damage to energy production infrastructure (due to wind) with difficulties to repair the network arising from high surface runoff/flood due to extreme rainfall. For this point (and the next one), a paragraph is added at the end of the introduction to justify the presented multi-hazard analysis and provide better context (P3 L76-84). • In the case of wildfire and extreme air temperature, we consider both of these hazards (UNISDR, 2009). Extreme hot temperature can lead to damage on infrastructure (e.g., rail track deformation) and cause death of people (e.g., 2003 heatwave in France). In term of impact of wildfire and extreme temperature, we argue that their interrelation

has the potential to put overwhelming pressure on critical infrastructure (e.g., health service, firefighters). That being said, this article is focusing on natural hazards and their interrelations and not on the measure of a combined impact, which can differ depending on the infrastructure/society considered. We hope that these explanations clarify the aims of the study. Again, to provide further clarification in the manuscript, we have added a paragraph at the end of the introduction to justify the presented multi-hazard analysis and provide better context (P3 L76-84).

[R2 Comment 2] 2. Wildfire burned area is related to temperature not the number of fires. A fire of size 0.1 ha can occur all year around, specially for human started fires (see Balch, J. K., et al. (2017). Human-started wildfires expand the fire niche across the United States. Proceedings of the National Academy of Sciences, 114(11), 2946-2951.). But more importantly, what is the impact? If it is the wildfire, is it justified to have fire as the impact and as the hazard? Is high-temperature necessarily a hazard?

[AR 2] We believe that extreme high temperatures can be considered a hazard based on sources such as UNISDR (2009). It is true that small fire occurs all year round, and our data also support that statement (see manuscript Figure 12). Burned area and number of wildfires depends on many other drivers such as wind, type of fuel and soil moisture. The aim of this study is not to decipher the mechanism leading to a wildfire but rather as an exemplar of looking at two hazards and quantifying their interrelation between two hazards. We have now made this point clearer in our manuscript. Regarding impact, both extreme high temperatures and wildfire as single hazards can potentially impact society. We also consider that extreme temperature increases the probability of wildfire occurring (AghaKouchak et al., 2018). We then try to address both of the following: (i) How much does an increase in tempearature impacts wildfire likelilhood; (ii) Estimating return periods for events of extreme heat and fire. A paragraph has been added at the beginning of Section 4.2 to justify our choices (P26 L452-457).

[R2 Comment 3] 3. Abstract needs some improvement. I struggled to understand how

the case studies are related to synthetic data.

[AR 3] This point has also been noted by Reviewer 1 (R1) and that part of the abstract has been modified (L10-11) and (L12-13)

[R2 Comment 4] 4. An important missing element in the evaluation of models is p-value. It would be interesting to see what the p-values are and determine whether the models fail/pass to represent the data! This is actually very important. The metrics used in the paper are subjective - although valuable - and a more widely accepted metric could help the general audience relate the study to other modelling practices.

[AR 4] We are unsure that p-value is applicable in the framework of this study as we are comparing curves and not the distributions themselves. However, we agree on the fact that a robust comparison point to assess the efficacity of model might be needed. We think that comparing model fitting capabilities to an empirical curve (obtained through point counting) would give valuable insight We therefore created empirical level curves and computed their wd to the reference curve following the same steps as for the 6 models. For each of the 60 synthetic datasets, models having a smaller wd to the reference curve than the empirical estimate are considered to be "passing" to represent the data with more accuracy than a naïve approach. Figure 7 has therefore been updated to incorporate this new information and a new paragraph has been added in Section 3.3 (P18 L484-489).

[R2 Comment 5] 5. There are many typos in the text. I highlighted some of them in the attached manuscript, but there are more.

[AR 5] Thank you for this noting of the typos. We have taken these into account in our revised manuscript by catching many of these, and will ensure that for any final proof reading we do a careful reading for typos.

[R2 Comment 6] 6. There are some specific comments that are provided in the at-tached manuscript. Hope the authors find the comments useful to improve their paper

[AR 6] Thank you for these comments. We have gone through these in detail and the major ones not addressed in R2 Comments 1 to 5 are the following, along with our replies:

[R2 Specific comments (from supplement)] L49-L51 P2 provide names not acronyms Change done

L 72 P3 joint extreme can be AND or OR scenario, Figure 1 shows AND scenario. please clarify It has been clarified

L108 P 4 Notations in equation 1&2 don't seem to be correctly presented Change done L112 P4 u merges to 1 not infinity (if u is marginal) Change done. Thank you!

L114 P4 respectively is not needed. It is confusing actually Change done

L142 P5 this "u" is never defined before It was a mistake. U has been replaced by z and z has been defined.

L170 P6 "and" not "or" Change done

L257 P6 Summary section not needed Section has been removed

L270 P6 is this the same "u" used before? if not, it will be confusing No, but the meaning of u is specified on the following line

Figure 7 P19 while the presented metrics are nice, it would be more desirable to show the p-value for each case, to examine whether the copula/any model is representative of the data, or not. Then the distances would help select the best model The figure has been redone and the "tiles" in which models have a lower wd than a naïve empirical approach (See AR4) is highlighted. We consider that if the model represents the data better than an empirical approach in one of the cases, then the model is suitable for that case.

L472 P20 How can you say that. How is wd<0.1 selected as the criterion for goodness-of-fit, why not 0.05 or 0.2? We modified this part and also use the fact that the wd of

the model is lower than the empirical wd as a criterion for goodness of fit (See AR4).

L488 P20 needs a comma here In general commas are missing throughout the manuscript. before "respectively" you need a comma, etc We have changed some of these in our revised manuscript, along with doing a more thorough proofreading.

L510 P20 comma after "here" Change done

L599 P25 not the right reference for wildfire. Suggestion: Littell, J. S., McKenzie, D., Peterson, D. L., & Westerling, A. L. (2009). Climate and wildfire area burned in western US ecoprovinces, 1916–2003. Ecological Applications, 19(4), 1003-1021. We added the reference, thank you.
* * *
[Figure]

**Fig. 1.** Weighted Normalized Euclidean Distance (wd) to the reference curve for all 60 different synthetic datasets. Fitting capacities of each model are represented. Values in cells and colours represent the

---

## Editor Decision (ED1)

[revised manuscript text omitted]